# Inconsistency-Aware Minimization: Improving Generalization with Unlabeled Data

## Abstract

Estimating the generalization gap and devising optimization methods that generalize better are crucial for deep learning models, both for theoretical understanding and for practical applications. The ability to leverage unlabeled data for these purposes offers significant advantages in real-world scenarios. This paper introduces a novel generalization measure, termed *local inconsistency*, developed from an information-geometric perspective of the neural network's parameter space. A key feature of local inconsistency is its computability from unlabeled data. We establish its theoretical underpinnings by connecting local inconsistency to the Fisher information matrix and the loss Hessian. Empirically, we demonstrate that local inconsistency correlates with the generalization gap also exhibiting characteristics comparable to *sharpness*. Based on these findings, we propose Inconsistency-Aware Minimization (IAM) that incorporates local inconsistency into the objective. We demonstrate that in standard supervised learning settings, IAM enhances generalization, achieving performance comparable to existing methods such as Sharpness-Aware Minimization. Furthermore, IAM exhibits efficacy in semi- and self-supervised learning scenarios, where the local inconsistency is computed from the unlabeled data.

## 1 Introduction

Estimating the generalization gap and developing optimization techniques to enhance performance on unseen data are pivotal challenges in both the theory and practice of deep learning. There are numerous reports linking the flatness of the loss landscape to generalization [16, 7, 20, 9]. Consequently, numerous minimization methods leveraging sharpness have been proposed, demonstrating improvements in generalization [8, 19, 17, 35] . However, the efficacy of such sharpness-based approaches is not fully understood as recent studies, including Andriushchenko et al. [1], indicate that sharpness alone does not reliably predict the generalization gap.

On the other hand, Jiang et al. [14], Johnson and Zhang [15] have investigated alternative generalization measures, such as *disagreement* and *inconsistency*. Under certain conditions (e.g., zero training error or low randomness of final states), these metrics have been shown to estimate the generalization gap more accurately than ones based on sharpness. From a practical standpoint, it is highly desirable to leverage unlabeled data to compute these measures. However, existing methods for computing disagreement typically require training two separate models under identical conditions (while still being subject to training randomness), and estimating inconsistency often necessitates training multiple models on each of distinct datasets, thereby incurring substantial computational overhead. Such prerequisites hinder the direct minimization of these measures within a single-model training paradigm.

Submitted to 39th Conference on Neural Information Processing Systems (NeurIPS 2025). Do not distribute.

In many real-world applications, unlabeled data is far more abundant and accessible than labeled data. Therefore, methods capable of estimating the generalization gap and subsequently minimizing it using only unlabeled data within a single-model framework (i.e., without resorting to auxiliary models or requiring held-out labels) are crucial for resource-efficient deployment in practical scenarios.

Addressing this need, this paper introduces *local inconsistency*, a novel generalization measure derived from an information-geometric perspective of the parameter space. This measure is theoretically grounded through its connection to the Fisher Information Matrix (FIM) and the loss Hessian. Crucially, local inconsistency can be computed using only **unlabeled** data and from a **single** trained model. We demonstrate that local inconsistency exhibits a predictive capability for the generalization gap comparable to sharpness-based measures in some settings. It also shows distinct characteristics where traditional sharpness measures may falter. Building upon local inconsistency, we propose **Inconsistency-Aware Minimization** (IAM), a novel optimization strategy that incorporates local inconsistency into the training objective. In standard supervised learning settings on CIFAR-10 and CIFAR-100[18], IAM achieves generalization performance comparable or superior to existing methods like Sharpness-Aware Minimization (SAM) [8]. Furthermore, IAM demonstrates its efficacy in semi- and self-supervised learning, achieving higher test accuracy than standard SGD when integrated with SimCLR [6].

Our contributions can be summarized as follows:

- We propose *local inconsistency*, a novel generalization measure rooted in information geometry, designed to capture the sensitivity of model outputs to perturbations in the parameter space. This measure offers significant practical advantages as it can be computed using (i) only unlabeled data and (ii) a single trained model.

- We establish the theoretical underpinnings of local inconsistency by linking it to the Fisher Information Matrix (FIM) and the loss Hessian. Furthermore, we discuss its approximate relationship with the inconsistency measure previously proposed by Johnson and Zhang [15].

- We develop Inconsistency-Aware Minimization (IAM), a novel optimization framework with two variants (IAM-D and IAM-S), that directly incorporates local inconsistency into the training objective to seek flatter minima in terms of output sensitivity. IAM demonstrates improved generalization performance across diverse learning paradigms, including standard supervised learning and self-supervised learning settings, showcasing its broad applicability.

## 2   Related work

Understanding and improving generalization in deep neural networks, especially given their large capacity and tendency to overfit [33], remains a central challenge. While networks can memorize random labels [33] and learn simple patterns before noise [2], phenomena like double descent [25] and the inadequacy of uniform convergence theory [24] highlight the need for novel generalization measures beyond loss-based metrics.

Traditional measures like VC-dimension often fall short. While spectrally-normalized margin bounds [3] and PAC-Bayes approaches offer insights, no single measure consistently predicts generalization [13]. Recently, disagreement [14] and inconsistency [15] have shown promise, correlating well with the generalization gap, even when computed on unlabeled data. However, their reliance on training multiple models poses practical limitations for direct optimization in a single-model setup, underscoring the need for efficient, label-free, single-model generalization measures.

The geometry of the loss landscape, particularly the flatness of minima, has been extensively linked to generalization [16, 20]. However, the utility of sharpness as a sole predictor is debated due to issues like scale invariance [7] and its correlation with training hyperparameters rather than true generalization [1]. Indeed, some studies suggest that output inconsistency and instability can be more reliable predictors than sharpness [15]. Information geometry has inspired reparametrization-invariant sharpness measure [12], but these can be computationally expensive. This context motivates our exploration of "local inconsistency", an alternative geometric measure focusing on output sensitivity within a parameter neighborhood, computable from unlabeled data using a single model.

Various regularization techniques, both explicit (e.g., dropout [30], batch normalization [28], Mixup [34]) and implicit (e.g., SGD's bias [10, 29]), aim to improve generalization. Methods like Sharpness-

Aware Minimization (SAM, [8]) and ASAM [19] directly optimize for flat minima and have shown significant improvements. Despite their success, the precise role of sharpness in generalization remains an active area of research [13, 1], further motivating the development of complementary approaches like our proposed IAM.

# 3 Background and preliminaries

In this section, we briefly review fundamental concepts and notations essential for understanding our proposed metric and its theoretical connections. We focus on probabilistic classification models, information geometry, and aspects of the loss landscape.

## 3.1 Notation and problem setup

We consider probabilistic classification models. Let $x \in \mathcal{X}$ be a data point from the input space $\mathcal{X}$, and $y \in [C] = \{0, 1, \ldots, C - 1\}$ be the corresponding class label, where $C$ is the total number of classes. The data pair $(x, y)$ are assumed to be drawn from an underlying distribution $\mathscr{D}$ over $\mathcal{X} \times [C]$. A model, parameterized by $\theta \in \mathbb{R}^m$, outputs a probability distribution over classes for a given input $x$. This is typically achieved by transforming a logit vector $z(x, \theta)$ through a softmax function: $f(x, \theta) = \mathrm{softmax}(z(x, \theta))$. Thus, $f(x, \theta) = [p(0|x; \theta), p(1|x; \theta), \ldots, p(C - 1|x; \theta)]^\top$. Given a training dataset $Z_n = \{(x_i, y_i) : i = 1, \ldots, n\}$ drawn i.i.d. from $\mathscr{D}$, the model is typically trained by minimizing a loss function. For classification, the empirical Cross-Entropy (CE) loss will be written as $L(\theta) = \frac{1}{n} \sum_{i=1}^{n} l_i(\theta)$, where per-sample loss is $l_i(\theta) = l(x_i, y_i; \theta) = -\log p(y_i|x_i; \theta)$.

## 3.2 Fisher information matrix (FIM) and KL divergence

The Fisher information matrix (FIM), $F(\theta)$, for the family of probability density $p(x, y; \theta) = p(x)p(y|x; \theta)$ parameterized by a parameters $\theta$ is defined as

$$\begin{aligned} F(\theta) &= \mathbb{E}_{x \sim p(x)} \left[ \mathbb{E}_{y \sim p(y|x;\theta)} \left[ \nabla_\theta l(x, y; \theta) \nabla_\theta l(x, y; \theta)^\top \right] \right] \\ &= \mathbb{E}_{x \sim p(x)} \left[ \nabla_\theta z(x_i, \theta) \left( \mathrm{diag}(f(x_i, \theta)) - f(x_i, \theta) f(x_i, \theta)^\top \right) \nabla_\theta z(x_i, \theta)^\top \right]. \end{aligned} \quad (1)$$

In practice, the expectation $\mathbb{E}_{p(x)}$ is often approximated by an empirical average over the available data (e.g., training data $\{x_i\}_{i=1}^n$ or unlabeled data).

The Kullback-Leibler (KL) divergence between the output distributions of a model with parameters $\theta$ and a slightly perturbed model $\theta + \delta$, $f(x, \theta)$ and $f(x, \theta + \delta)$, respectively, can be locally approximated using a second-order Taylor expansion with respect to $\delta$ as:

$$\mathbb{E}_{x \sim p(x)} \left[ \mathrm{KL} \left( f(x, \theta) \| f(x, \theta + \delta) \right) \right] = \frac{1}{2} \delta^\top F(\theta) \delta + O(\|\delta\|_2^3). \quad (2)$$

## 3.3 Loss Hessian and Gauss-Newton approximation

The geometry of the empirical loss surface $L(\theta)$ is described by its Hessian matrix $H(\theta) = \nabla_\theta^2 L(\theta)$. For the Cross-Entropy (CE) loss, the Hessian can be approximated by the Gauss-Newton (GN) matrix, $G(\theta)$. The second derivative of the per-sample CE loss $\ell_i(\theta)$ with respect to the logits $z_i = z(x_i, \theta)$, $\nabla_z^2 \ell_i(\theta) = \mathrm{diag}(f(x_i, \theta)) - f(x_i, \theta) f(x_i, \theta)^\top$, depends only on the model's output probabilities $f(x_i, \theta)$. Consequently, the per-sample GN term, $G_i(\theta) = \nabla_\theta z_i^\top (\nabla_z^2 \ell_i) \nabla_\theta z_i$, is equivalent to the FIM contribution in Eq. (1). The empirical GN matrix, $G(\theta) = \frac{1}{n} \sum_{i=1}^{n} G_i(\theta)$, thus often termed the empirical FIM, provides a positive semi-definite approximation to $H(\theta)$ and is frequently used in optimization [22, 26].

# 4 Accessing generalization gap via local inconsistency

This section introduces our proposed measure, local inconsistency, designed to capture the generalization gap. We first define local inconsistency and elucidate its theoretical underpinnings by connecting it to the FIM and the loss Hessian. We then discuss its relationship with inconsistency [15]. Finally, we present empirical results demonstrating the correlation between local inconsistency and the generalization gap, comparing it with other common measures.

### 4.1 Local inconsistency, $S_\rho(\theta)$

We introduce local inconsistency, $S_\rho(\theta)$, defined as:

$$S_\rho(\theta) = \max_{\|\delta\| \leq \rho} \mathbb{E}_{x \sim p(x)}[\mathrm{KL}(f(x,\theta)\|f(x,\theta+\delta))], \tag{3}$$

which represents the sensitivity of the model's output distribution $f(x,\theta)$ with respect to the worst perturbations $\delta$, within an Euclidean ball of radius $\rho$ around the parameter $\theta$. Intuitively, a high value of $S_\rho(\theta)$ indicates that the model's output distribution is highly sensitive to small perturbations in parameter space. This sensitivity suggests potential instability or uncertainty in the model's predictions associated with the vicinity of $\theta$.

**Practical Advantages of $S_\rho$** Local inconsistency shares a practical advantage with sharpness-based measures [16, 8] in that it can be calculated using a **single** trained model. Furthermore, like disagreement [14] and inconsistency [15], our metric can be estimated using only **unlabeled** data. A notable advantage over inconsistency and disagreement estimation is that evaluating $S_\rho$ does not require training multiple model instances derived from the same training procedure. This potentially makes $S_\rho$ more computationally efficient and practical to compute, especially when model training is resource-intensive.

### 4.2 Connection to FIM and Hessian

The relationship between our metric $S_\rho$ and the Fisher Information Matrix (FIM) can be established by leveraging the local quadratic approximation of the KL divergence, as outlined in Section 3. With this quadratic approximation, we can approximate $S_\rho(w)$ with the maximum eigenvalue of FIM, scaled by $\rho^2/2$:

$$S_\rho(\theta) \approx \max_{\|\delta\| \leq \rho} \frac{1}{2}\delta^\top F(\theta)\delta = \frac{1}{2}(\rho v_{\max})^\top F(\theta)(\rho v_{\max}) = \frac{1}{2}\rho^2 \lambda_{\max},$$

where $v_{\max}$ is the eigenvector corresponding to the largest eigenvalue $\lambda_{\max}$ of $F(\theta)$. Remarkably, this approximation requires only the model $\theta$ and unlabeled data (used to compute the expectation).

The Fisher Information Matrix $F(\theta)$, to which $S_\rho(\theta)$ is related via its maximum eigenvalue, also connects to the Hessian of the loss function $H(\theta)$. As detailed in Section 3, for Negative Log Likelihood losses such as Cross-Entropy, the Hessian can be approximated by the Gauss-Newton matrix $G(\theta)$, equivalent to empirical FIM computed using training data.

Consequently, when calculating $S_\rho(\theta)$ using the training data, it approximates $\frac{1}{2}\rho^2 \lambda_{\max}(G(\theta))$. Given that $G(\theta)$ often provides a good approximation to the true loss Hessian near a local minimum, $S_\rho(\theta)$ therefore offers insights into the maximum curvature of the loss landscape in that vicinity.

### 4.3 Local Inconsistency and Generalization Bound

Beyond its connection to the local geometry of the loss landscape, our proposed local inconsistency measure, $S_\rho(\theta)$, can also be linked to the generalization ability of the model. Inspired by PAC-Bayesian analyses that connect the geometry of the loss neighborhood to generalization [8], we can sketch a theoretical argument suggesting that controlling $S_\rho(\theta)$ contributes to a generalization bound. We present an informal theorem that captures this intuition, with the detailed heuristic derivation provided in Appendix C.

**Theorem 1 (Informal Generalization Bound with Local Inconsistency)** *Under certain assumptions regarding the relationship between the worst-case empirical loss increase in a $\rho$-neighborhood and the local inconsistency $S_\rho(\theta)$ (evaluated on the training set $Z_n$), the true risk $L_{\mathscr{D}}(\theta) = \mathbb{E}_{(x,y)\sim\mathscr{D}}[l(x,y;\theta)]$ can be bounded with high probability as:*

$$L_{\mathscr{D}}(\theta) \lesssim L(\theta) + S_\rho(\theta) + \mathcal{R}(\|\theta\|^2/\rho^2) \tag{4}$$

*where $\mathcal{R} : \mathbb{R} \to \mathbb{R}^+$ is a strictly increasing function.*

This bound (Eq. (4)) suggests that minimizing a combination of the empirical loss $L_\mathcal{S}(\theta)$ and the local inconsistency $S_\rho(\theta)$ can lead to a lower upper bound on the true risk. This provides a theoretical motivation for our Inconsistency-Aware Minimization (IAM) framework, which aims to find solutions that are not only accurate on the training data but also exhibit low output sensitivity in the parameter space, as measured by $S_\rho(\theta)$.

## 4.4 Relation with inconsistency in Johnson and Zhang [15]

Local inconsistency exhibits an interesting relationship to the inconsistency in Johnson and Zhang [15] defined as:

$$\mathcal{C}_P = \mathbb{E}_{Z_n}\mathbb{E}_{\theta,\theta'\sim\Theta_{P|Z_n}}\mathbb{E}_{x\sim p(x)}[\mathrm{KL}(f(x,\theta)\|f(x,\theta'))].$$

We consider the conditional inconsistency for a fixed $Z_n$, denoted $\mathcal{C}_{P|Z_n}$, without outer expectation. Then our proposed metric, $S_\rho(\theta_{Z_n})$, is approximately proportional to the conditional inconsistency $\mathcal{C}_{P|Z_n}$:

$$\frac{m}{2C}\mathcal{C}_{P|Z_n} \lesssim S_\rho(\theta_{Z_n}) \lesssim \frac{m}{2}\mathcal{C}_{P|Z_n}, \tag{5}$$

under certain assumptions, such as assuming the parameter posterior $\Theta_{P|Z_n}$ as a distribution with isotropic covariance and $\theta_{Z_n}$ as mean. This connection arises because both metrics are related to the local geometry captured by the FIM at $\theta_{Z_n}$, with $S_\rho$ being linked to its maximum eigenvalue and $\mathcal{C}_{P|Z_n}$ to its trace. Practically, the eigenspectra of the FIM of a neural network are observed to be dominated by a few large eigenvalues (specifically related to the number of classes, $C$ in classification task) while remaining eigenvalues are near zero. This observation indicates that the ratio $\lambda_{\max}(F(\theta))/\mathrm{Tr}(F(\theta))$ is larger than $\frac{1}{C}$ ($C \ll m$). For detailed derivation, please see Appendix B.

## 4.5 Estimating $S_\rho(w)$

Directly computing $S_\rho(w)$ requires solving the maximization problem over the high-dimensional parameter perturbation $\delta$. For deep neural networks, finding the exact maximum within the $L_2$-ball of radius $\rho$ is generally intractable. Therefore, we resort to numerical approximation methods.

For small perturbations $\delta$, the expected KL divergence can be accurately approximated by a second-order Taylor expansion involving the Fisher Information Matrix (FIM), $F(\theta)$, as Eq. (2) in Section 3 . Under quadratic approximation, as discussed in Section 4.2, the optimal perturbation $\delta^* = \rho v_{\max}$, the maximum value is then $S_\rho(\theta) = \frac{1}{2}\rho^2 \lambda_{\max}$, and the gradient of the approximated KL divergence with respect to $\delta$ is $F(\theta)\delta$.

This connection motivates not an usual Projected Gradient Ascent, that update $\delta_{k+1} \leftarrow \Pi_{\{\delta_k:\|\delta_k\|\leq\rho\}}(\delta_k + \eta F(\theta)\delta_k)$, but an iterative gradient ascent approach that update

$$\delta_{k+1} = \frac{\rho}{\|F(w)\delta_k\|}F(w)\delta_k, \qquad \delta_0 = \varepsilon \sim \mathcal{N}\left(0, \frac{\sigma^2}{m}I_m\right),$$

where $\sigma^2$ is initial noise scale. Iterative gradient ascent is precisely one iteration of the Power Iteration method used to find the dominant eigenvector of $F(w)$.

---

**Algorithm 1** Estimation of $S_\rho(w)$

1: **Input:** model parameter $w \in \mathbb{R}^m$, noise scale $\sigma^2$,
2: radius $\rho > 0$, number of steps $K \geq 1$
3: **Initialize** $\delta_0$ randomly with $\mathcal{N}(0, \frac{\sigma^2}{m}I_m)$
4: **for** $k = 0$ to $K - 1$ **do**
5:     Compute $g_k = \nabla_\delta\mathbb{E}_{x\sim p(x)}\mathrm{KL}(f(x,\theta)\|f(x,\theta+\delta))|_{\delta=\delta_k}$
6:     Update perturbation: $\delta_{k+1} = \rho\frac{g_k}{\|g_k\|_2}$
7: **end for**
8: **return** $\mathbb{E}_{x\sim p(x)}\mathrm{KL}(f(x,\theta)\|f(x,\theta+\delta_K))$

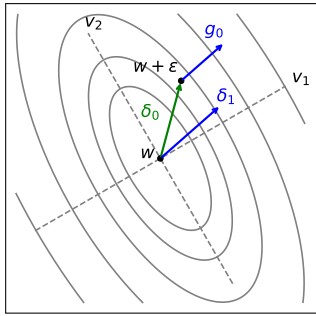

## 4.5.1 Algorithm for estimating $S_\rho(w)$

Based on the above, we propose Algorithm 1 to estimate $S_\rho(w)$. This algorithm performs $K$ steps of normalized gradient ascent (effectively, Power Iteration under the quadratic approximation) to find an approximate maximizing perturbation $\delta^*$.

### 4.6 Empirical results

To assess the predictive capability of local inconsistency $S_\rho$ for the generalization gap, we conducted experiments on CIFAR-10. We trained two distinct architectures, a 6-layer CNN (6CNN) and a Wide Residual Network (WRN28-2)[32], under various hyperparameter settings (details in Appendix E). $S_\rho$ was estimated using a disjoint, unlabeled data set. For comparison, we also computed two common sharpness-based measures: the trace, $\mathrm{Tr}(H)$, and the maximum eigenvalue, $\lambda_{\max}(H)$.

Figure 1 presents scatter plots of these metrics against the generalization gap, with Kendall's Tau ($\tau$) reported for each. For the simpler 6CNN model (top row), $S_\rho$ ($\tau = 0.5141$) exhibited a positive correlation with the generalization gap, comparable to $\mathrm{Tr}(H)$ ($\tau = 0.5444$) and $\lambda_{\max}(H)$ ($\tau = 0.5175$). This suggests that for smaller models, various geometric measures may similarly capture aspects of generalization. However, for the larger WRN28-2 model with data augmentation (bottom row), a more nuanced behavior emerged. As noted by Andriushchenko et al. [1], different training configurations can form distinct solution subgroups. In our WRN28-2 experiments, $\mathrm{Tr}(H)$ and $\lambda_{\max}(H)$ showed positive correlations only within such subgroups, but exhibited negative overall correlations globally ($\tau = -0.0439$ and $\tau = -0.1200$, respectively). In stark contrast, our $S_\rho$ maintained a positive, albeit reduced, correlation across all settings ($\tau = 0.3658$).

This divergence, particularly with larger models and data augmentation, suggests that local inconsistency captures information about the generalization gap that is distinct from, or complementary to, traditional Hessian-based sharpness. While the predictive utility of sharpness metrics can be confounded by these subgroup effects, $S_\rho$ demonstrates more consistent global predictiveness, hinting at its potential as a more robust generalization indicator in complex training scenarios.

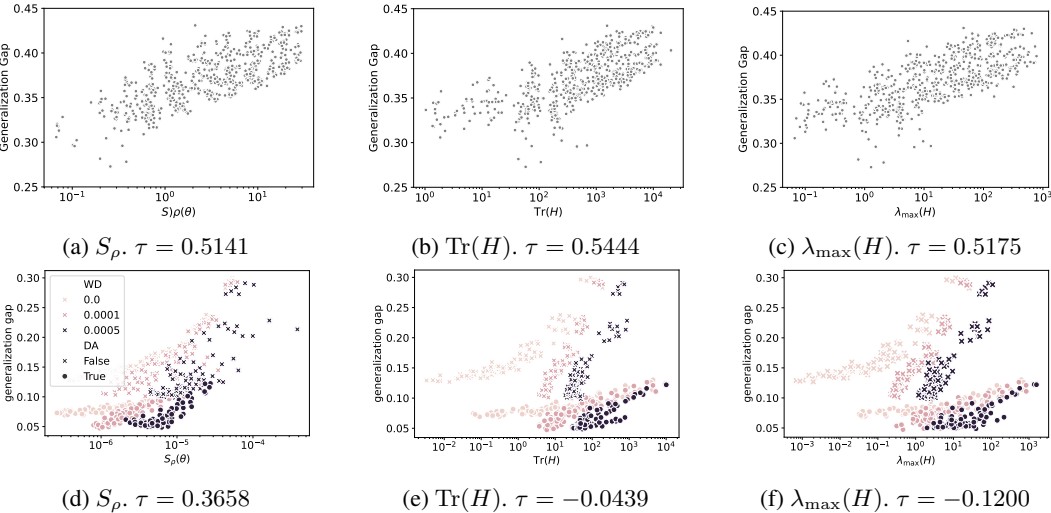

Figure 1: Local inconsistency and sharpness measures vs the generalization gap.

## 5 Inconsistency-Aware Minimization (IAM): incorporating local inconsistency into the objective

Our empirical findings suggest that local inconsistency, $S_\rho(\theta)$ defined in Eq. (3), correlates with the generalization gap. This motivates its use as a regularizer to guide the optimization towards solutions that not only fit the training data, but also exhibit low sensitivity in their output distributions with respect to parameter perturbations. We propose two strategies to incorporate local inconsistency into the training objective.

1. **Direct Regularization (IAM-D)**: This approach directly penalizes local inconsistency by adding it to the standard training loss $L(\theta)$:

$$L_{\text{IAM-D}}(\theta) = L(\theta) + \beta S_\rho(\theta) = L(\theta) + \beta \max_{\|\delta\|_2 \le \rho} \mathbb{E}_X[\mathrm{KL}(f(X, \theta) \| f(X, \theta + \delta))], \qquad (6)$$

where $\beta > 0$ is a hyperparameter balancing the trade-off. This objective seeks parameter values $\theta$ for which the model outputs are robust across the neighborhood defined by $\rho$.

2. **SAM-like Approach (IAM-S)**: Inspired by SAM [8], this method aims to find parameters $\theta$ that reside in a neighborhood of uniformly low loss by minimizing the loss at an adversarially perturbed point $\theta + \delta^*$:

$$L_{\text{IAM-S}}(\theta) = L(\theta + \delta^*), \quad \text{where } \delta^* = \underset{\|\delta\|_2 \le \rho}{\arg\max} \, \mathbb{E}_X[\text{KL}(f(X, \theta)\|f(X, \theta + \delta))]. \quad (7)$$

Here, $\delta^*$ is the perturbation that maximizes the local inconsistency term. Note that the objective minimizes the original loss $L$ at the perturbed point $\theta + \delta$:

$$L(\theta + \delta) \approx L(\theta) + \delta^\top \nabla_\theta L(\theta) + \frac{1}{2}\delta^\top G(\theta)\delta.$$

Thus, IAM-S implicitly minimizes the principal eigenvalues of $G(\theta)$, equivalent to empirical FIM.

In the following subsections, we detail the algorithm for IAM-S and provide an analysis of its objective. The algorithm for IAM-D involves a similar inner maximization for $S_\rho(\theta)$ followed by a standard gradient descent step on $L_{\text{IAM-D}}(\theta)$.

## 5.1 Algorithm for IAM-D and IAM-S

Optimizing $L_{\text{IAM-S}}(\theta)$ and $L_{\text{IAM-S}}(\theta)$ involves a min-max procedure. The inner maximization to find $\delta^*$ (i.e., computing $S_\rho(\theta)$ and the corresponding $\delta^*$) is performed using an Algorithm 1, typically for $K = 1$ step for efficiency. IAM-D simply add the $\beta S_\rho(\theta)$ with $\delta_K$ to the $L(\theta)$, and then update $\theta$ with standard SGD. The outer minimization step of IAM-S updates $\theta$ based on the gradient of the loss $L(\theta + \delta_K)$ dropping the second-order terms same with SAM: $\nabla_\theta L_{\text{IAM-S}}(\theta) \approx \nabla_\theta L(\theta)|_{\theta=\theta+\delta_K}$. This two-step process is summarized in Algorithm 2 in Appendix D.

## 5.2 Empirical evaluation in supervised learning

We evaluated the performance of IAM against SGD and SAM in image classification tasks. WRN16-8[32] served as the baseline model, trained on CIFAR-10, 100 with basic augmentations. Optimal hyperparameters for IAM-D were found to be $\beta = 1.0, \rho = 0.1$ for CIFAR-10, and $\beta = 10.0, \rho = 0.1$ for CIFAR-100, and for IAM-S were $\rho = 0.1, 0.5$ in CIFAR-10 and CIFAR-100 respectively. Table 1 summarizes the test error rates. Both IAM-D (Direct Regularization) and IAM-S (SAM-like Approach) variants not only reduce test error compared to SGD but also achieve performance comparable to SAM. Notably, on CIFAR-100, IAM-S outperforms SAM by a margin of 0.75%, demonstrating its effectiveness in more complex datasets.

Table 1: Test Error (mean $\pm$ stderr) of IAM, SAM, and SGD on WRN-16-8 trained with CIFAR-10, CIFAR-100

|  | SGD | SAM | IAM-D | IAM-S |
|---|---|---|---|---|
| CIFAR-10 | $3.95_{\pm 0.048}$ | $3.31_{\pm 0.010}$ | $\mathbf{3.28}_{\pm 0.060}$ | $3.30_{\pm 0.042}$ |
| CIFAR-100 | $19.17_{\pm 0.192}$ | $17.63_{\pm 0.119}$ | $17.16_{\pm 0.028}$ | $\mathbf{16.88}_{\pm 0.021}$ |

Figure 2 illustrates the evolution of local inconsistency $S_\rho(\theta)$ and test accuracy during training for SGD and IAM-D. IAM-D effectively suppresses the increase in $S_\rho(\theta)$ and mitigates overfitting, particularly evident after learning rate decay points where test accuracy for SGD can degrade. Both on CIFAR-10, 100 (Figure 2), IAM-D maintains $S_\rho(\theta)$ below SGD. Although second LR decay temporarily reduces inconsistency for both, SGD's inconsistency quickly rebounds, unlike the stable behavior of IAM-D. These observations suggest that minimizing local inconsistency helps confine the model to parameter regions with smoother output distributions, correlating with the generalization improvements shown in Table 1.

## 5.3 IAM for Learning with Limited or No Explicit Labels

A key advantage of local inconsistency is its computability from unlabeled data, making IAM well-suited for scenarios with limited or no explicit supervision. We demonstrate this in semi-supervised and self-supervised learning settings. Detailed settings are listed in Appendix E.

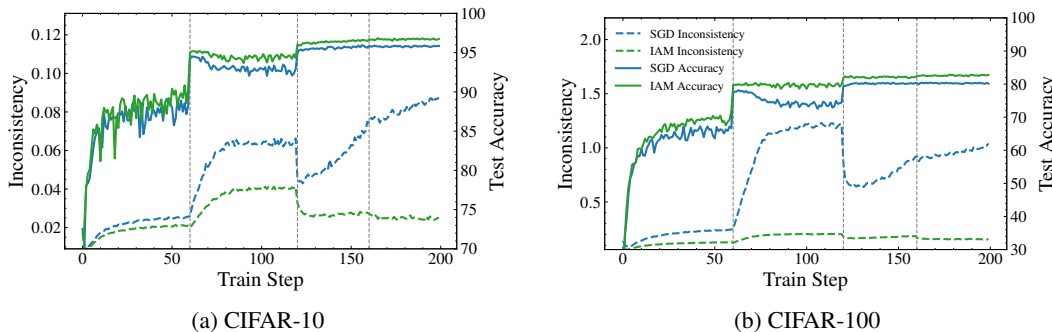

(a) CIFAR-10         (b) CIFAR-100

Figure 2: The evolution of the local inconsistency $S_\rho(\theta)$ and test accuracy with SGD and IAM-D.

**Semi-Supervised Learning.** In many practical scenarios, labeled data is scarce while unlabeled data is abundant. We simulated semi-supervised learning on CIFAR-10 and CIFAR-100 by masking 80% to 99% of training labels. IAM-D was configured to optimize a joint objective: cross-entropy loss on the labeled subset plus the local inconsistency penalty computed over the entire mini-batch (both labeled and unlabeled examples). In contrast, SGD and SAM utilized only the labeled examples. As summarized in Table 2, IAM-D consistently outperforms both baselines across most missing-label ratios, confirming that leveraging unlabeled data via the local inconsistency term enhances robustness when supervision is sparse.

Table 2: Test Error (mean $\pm$ stderr) of IAM-D, SAM, and SGD on WRN-16-8 trained with CIFAR-10 and CIFAR-100 with varying label rates.

| Dataset | Model | Label Rate | | | |
|---------|-------|------|------|------|------|
| | | 1% | 5% | 10% | 20% |
| CIFAR-10 | SGD | $56.54 \pm 0.159$ | $28.64 \pm 1.648$ | $21.93 \pm 0.234$ | $17.42 \pm 0.430$ |
| | SAM | $55.83 \pm 0.728$ | $28.45 \pm 0.119$ | $20.68 \pm 0.102$ | $\mathbf{14.00} \pm 0.075$ |
| | IAM-D | $\mathbf{52.78} \pm 0.497$ | $\mathbf{25.66} \pm 0.723$ | $\mathbf{19.44} \pm 0.354$ | $14.24 \pm 0.142$ |
| CIFAR-100 | SGD | $89.35 \pm 0.098$ | $72.65 \pm 0.519$ | $60.86 \pm 0.204$ | $50.01 \pm 0.299$ |
| | SAM | $89.31 \pm 0.156$ | $72.02 \pm 0.337$ | $58.04 \pm 0.594$ | $45.64 \pm 0.085$ |
| | IAM-D | $\mathbf{88.36} \pm 0.292$ | $\mathbf{69.99} \pm 0.649$ | $\mathbf{57.60} \pm 0.251$ | $\mathbf{45.05} \pm 0.956$ |

**Self-Supervised Learning (SSL).** The label-agnostic nature of IAM makes it directly applicable to SSL objectives. We integrated IAM-D into the SimCLR framework [6], training a ResNet-18[11] encoder on CIFAR-10. Performance was evaluated via linear probing. The local inconsistency term for IAM-D was computed using the model's projection-head outputs. Figure 3 shows that SimCLR trained with IAM-D (SimCLR-IAM) achieves higher test accuracy on the downstream linear classification task compared to vanilla SimCLR (SimCLR-SGD). Furthermore, SimCLR-IAM tends to converge faster in terms of test error and also minimizes the SimCLR training loss more rapidly, despite the additional local inconsistency regularization. This suggests that controlling local inconsistency is beneficial even when no explicit labels are available during representation learning.

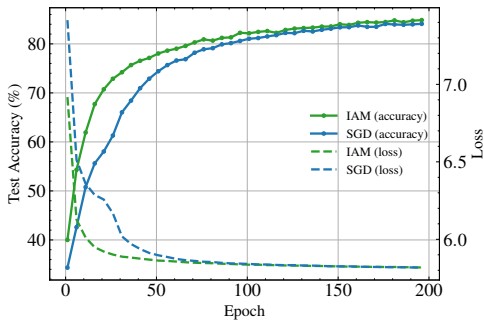

Figure 3: Test accuracy on linear probe and SimCLR training loss for ResNet-18 on CIFAR-10, comparing SimCLR trained with SGD (SimCLR-SGD) versus SimCLR with IAM-D (SimCLR-IAM).

### 5.4 IAM as implicit output entropy regularization

Improving generalization in deep neural networks often involves modulating the entropy of model output distributions. Techniques such as Label Smoothing (LS) [31, 23, 4] and Entropy Regularization (ER) [27, 5, 34] mitigate the common issue of overconfidence of cross-entropy loss minimization, by specifically targeting this output entropy.

Our Inconsistency-Aware Minimization (IAM) framework regularizes local inconsistency, $S_\rho(\theta)$. To understand its regularizing effect, we can leverage the identity $\text{KL}(P\|Q) = -H(P) + \text{CE}(P, Q)$, where $H(P)$ is the entropy of P and $\text{CE}(P, Q)$ is cross entropy of $P$ and $Q$. Since the expectation is linear and $H(f(x; \theta))$ does not depend on $\delta$, we can rewrite $S_\rho$ as:

$$S_\rho(\theta) = -\mathbb{E}_{x \sim p(x)}[H(f(x, \theta))] + \max_{\|\delta\|_2 \leq \rho} \mathbb{E}_{x \sim p(x)}[\text{CE}(f(x, \theta), f(x, \theta + \delta))]. \tag{8}$$

Minimizing $S_\rho(\theta)$ as defined in Eq. (8) thus involves two concurrent objectives. First, maximizing the expected output entropy $\mathbb{E}_X[H(f(x, \theta))]$. This discourages overconfident (low-entropy) predictions. Second, Minimizing the worst-case expected cross-entropy between the original model's output $f(x, \theta)$ and the perturbed model's output $f(x, \theta + \delta)$, which promotes stability of the output distribution under parameter perturbations. This dual objective resonates with the goals of established regularization techniques like LS and ER. LS effectively increases the entropy of the target distribution, thereby preventing the model from becoming overly confident in its predictions for the training labels. ER directly penalizes the low-entropy output distributions of the model. Formally, LS can be interpreted as minimizing a KL divergence to a smoothed target (e.g., $\text{KL}(q_{LS}(y|x)\|p(y|x; \theta))$), while ER often involves minimizing $\text{KL}(p(y|x; \theta)\|u)$, while $u$ is a uniform distribution, both encouraging the model to avoid excessive output certainty.

The first term in Eq. (8) ($-\mathbb{E}_X[H(f(x, \theta))]$) directly aligns with the aim of ER to penalize low-entropy outputs. The second term, by minimizing the "distance" (via CE) to the output of a perturbed model, enforces a form of local distributional stability. If highly confident predictions are less stable in parameter perturbation, then minimizing $S_\rho(\theta)$ would implicitly penalize such an overconfidence.

Therefore, minimizing local inconsistency $S_\rho(\theta)$ acts as a regularization strategy that not only promotes prediction stability against minor parameter variations but also indirectly encourages higher output entropy, akin to mechanisms in LS and ER. This multifaceted regularization is anticipated to yield improved generalization by encouraging the model to learn more robust and less overconfident representations, making them less susceptible to training data idiosyncrasies or parameter instabilities [34, 8].

## 6 Conclusion

In this work, we introduced "local inconsistency," a novel information-geometric generalization measure computable from a single model using only unlabeled data. We theoretically linked it to the Fisher Information Matrix (FIM) and the loss Hessian. Empirically, local inconsistency correlates with the generalization gap and exhibits distinct characteristics from traditional sharpness-based metrics.

Based on this, we proposed Inconsistency-Aware Minimization (IAM), an optimization framework that directly incorporates local inconsistency into the training objective. IAM enhances generalization in supervised learning, matching or exceeding that of Sharpness-Aware Minimization (SAM). Crucially, IAM proves effective in semi- and self-supervised learning by leveraging unlabeled data for local inconsistency computation, improving performance in label-scarce settings. We also elucidated IAM's mechanism as an implicit regularizer of model output entropy.

These findings offer a practical and theoretically-grounded approach to improving model generalization, particularly valuable in real-world applications where labeled data is limited. Future research could focus on a more rigorous establishment of the theoretical relationship between local inconsistency and generalization bounds (such as the informal bound presented in Theorem 1) and on exploring the scalability and applicability of IAM to a wider array of model architectures and large-scale datasets.

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

## A Theoretical Analysis: Generalization Bound with Local Inconsistency

In this section, we provide a theoretical sketch to connect our proposed local inconsistency measure, $S_\rho(\theta)$, to the generalization error of a model. Our goal is to show that minimizing $S_\rho(\theta)$, along with the empirical loss, can lead to a tighter upper bound on the true risk, thereby offering a theoretical motivation for our Inconsistency-Aware Minimization (IAM) approach We adapt insights from PAC-Bayesian theory, particularly drawing from the analysis of Sharpness-Aware Minimization (SAM) [8].

Let $L_{\mathscr{D}}(\theta) = \mathbb{E}_{(X,Y)\sim\mathscr{D}}[l(f(X,\theta),Y)]$ be the true risk and $L_{\mathcal{S}}(\theta) = \frac{1}{n}\sum_{i=1}^{n} l(f(X_i,\theta),Y_i)$ be the empirical risk on a training set $\mathcal{S}$ of size $n$. Foret et al. [8] provide a PAC-Bayesian generalization bound (informally stated, see their Appendix A for the full theorem [Thm 1, 8]) which, with high probability $1-\xi$ over the draw of $\mathcal{S}$, states:

$$L_{\mathscr{D}}(\theta) \leq \max_{\|\epsilon\|_2 \leq \rho} L_{\mathcal{S}}(\theta+\epsilon) + \mathcal{R}(\theta,\rho,n,m,\xi) \tag{9}$$

where $\mathcal{R}(\theta,\rho,n,m,\xi)$ is a complexity term that depends on the norm of the parameters $\|\theta\|_2^2$, the perturbation radius $\rho$, the number of training samples $n$, the number of parameters $m$, and the confidence $\xi$. Specifically,

$$\mathcal{R}(\theta,\rho,n,m,\xi) = \sqrt{\frac{m\log\left(1+\frac{\|\theta\|_2^2}{\rho^2}\left(1+\sqrt{\frac{\log n}{m}}\right)^2\right)+4\log\frac{n}{\xi}+\tilde{O}(1)}{n-1}}.$$

The first term on the right-hand side of Eq. (9) can be rewritten as:

$$\max_{\|\epsilon\|_2 \leq \rho} L_{\mathcal{S}}(\theta+\epsilon) = L_{\mathcal{S}}(\theta) + \left[\max_{\|\epsilon\|_2 \leq \rho}(L_{\mathcal{S}}(\theta+\epsilon)-L_{\mathcal{S}}(\theta))\right]. \tag{10}$$

The term in the square brackets, let's call it $\mathcal{SHARP}_\rho(\theta)$, represents the sharpness of the loss landscape at $\theta$ within a $\rho$-neighborhood, as considered by SAM. Our aim is to relate this sharpness term to our proposed local inconsistency measure $S_\rho(\theta)$.

Recall our definition of local inconsistency (calculated with respect to the empirical distribution over the training set $\mathcal{S}$ for this analysis):

$$S_\rho(\theta) = \max_{\|\delta\|_2 \leq \rho} \mathbb{E}_{X\sim\mathcal{S}}[\mathrm{KL}(f(X,\theta)\|f(X,\theta+\delta))]. \tag{11}$$

For the cross-entropy loss $l(f(X,\theta),Y) = -\log f(X,\theta)_Y$, under certain conditions, particularly near a good minimizer where the model's predictions $f(X,\theta)$ are close to the true label distribution (or if we consider $f(X,\theta)$ as a reference distribution), the change in loss due to parameter perturbation can be related to the KL divergence of the output distributions. Specifically, for a single sample $(X,Y)$, a second-order Taylor expansion of the loss $l(f(X,\theta+\delta),Y)$ around $\delta=0$ gives:

$$L_S(\theta+\delta) - L_S(\theta) = \nabla_\theta L_S(\theta)^\top \delta + \frac{1}{2}\delta^\top \nabla_\theta^2 L_S(\theta)\delta + O(\|\delta\|^3). \tag{12}$$

If $\theta$ is a point where $\nabla_\theta L_{\mathcal{S}}(\theta) \approx 0$ (i.e., near a minimum of the empirical risk), the first-order term in the empirical average $L_{\mathcal{S}}(\theta+\delta) - L_{\mathcal{S}}(\theta)$ becomes small. Furthermore, as discussed in Section 4.2 (and your Background section), for cross-entropy loss, the Hessian $\nabla_\theta^2 L_{\mathcal{S}}(\theta)$ can be approximated by the empirical Fisher Information Matrix (or Gauss-Newton matrix) $F_{\mathcal{S}}(\theta)$. Also, we know that $\mathbb{E}_{X\sim\mathcal{S}}[\mathrm{KL}(f(X,\theta)\|f(X,\theta+\delta))] \approx \frac{1}{2}\delta^\top F_{\mathcal{S}}(\theta)\delta$ for small $\delta$.

**Assumption (Approximate Equivalence of Loss Increase and Output KL-Divergence):** We heuristically assume that, for well-trained models near a local minimum, the worst-case increase in empirical loss due to parameter perturbation is approximately proportional to the local inconsistency (our $S_\rho(\theta)$ defined on the empirical data $\mathcal{S}$):

$$\max_{\|\epsilon\|_2 \leq \rho}(L_{\mathcal{S}}(\theta+\epsilon)-L_{\mathcal{S}}(\theta)) \approx c \cdot S_\rho(\theta) \tag{13}$$

for some positive constant $c$, which may depend on factors like the temperature of a Gibbs distribution if one were to formally link the loss to KL divergence from a Bayesian perspective (e.g., $c=1$ if the

loss essentially measures KL divergence to empirical targets). This assumption relies on the idea that changes in model output distributions (measured by KL divergence) are primary drivers of changes in the CE loss, especially for worst-case perturbations. While this is an approximation, it captures the intuition that models whose outputs are highly sensitive to parameter changes (high $S_\rho(\theta)$) are likely to experience larger increases in loss under such perturbations.

Substituting Eq. (13) into Eq. (10), and then into Eq. (9), we obtain the following generalization bound:

$$\boxed{L_{\mathscr{D}}(\theta) \lesssim L_{\mathcal{S}}(\theta) + c \cdot S_\rho(\theta) + \mathcal{R}(\theta, \rho, n, m, \xi)} \tag{14}$$

where $\lesssim$ indicates that the inequality relies on the approximation in Eq. (13).

**Interpretation and Implications.** The bound in Eq. (14) suggests that the true risk $L_{\mathscr{D}}(\theta)$ is upper-bounded by the sum of the empirical risk $L_{\mathcal{S}}(\theta)$, our local inconsistency measure $S_\rho(\theta)$ (scaled by a constant $c$), and the PAC-Bayesian complexity term $\mathcal{R}$. This provides a theoretical rationale for our IAM procedure: by minimizing an objective that includes both the empirical loss and the local inconsistency $S_\rho(\theta)$ (as in IAM-Direct, or implicitly by IAM-S seeking regions where $S_\rho(\theta)$ allows $L_{\mathcal{S}}(\theta + \delta^*)$ to be low), we are effectively attempting to minimize this upper bound on the true risk. A smaller $S_\rho(\theta)$ contributes to a tighter bound, potentially leading to better generalization.

The constant $c$ and the tightness of the approximation in Eq. (13) warrant further investigation. However, this sketch provides a plausible pathway to connect $S_\rho(\theta)$ with generalization guarantees by leveraging existing PAC-Bayesian frameworks that deal with loss landscape geometry. Rigorously establishing the relationship in Eq. (13) or deriving a similar bound with $S_\rho(\theta)$ appearing more directly through the KL term in a PAC-Bayes analysis (perhaps by defining a posterior whose "spread" is related to $S_\rho(\theta)$) are important directions for future theoretical work.

# B  Relation between our metric and inconsistency

This section outlines an approximate derivation relating the model output inconsistency $\mathcal{C}_P$, as defined by Johnson and Zhang [15], to the local sensitivity metric $S_\rho(w)$ defined previously. we will show simple demonstrations that these two metrics are related primarily through the Fisher Information Matrix (FIM), under specific assumptions like isotropic covariance. Then will show results with anisotropic covariance.

**Definitions**

- **Inconsistency** ($\mathcal{C}_P$): Measures the average difference (in terms of KL divergence) between the outputs of models generated by a stochastic training procedure $P$ applied to the same training data $Z_n$. The average is taken over draws of the training data $Z_n$ and pairs of models $(\Theta, \Theta')$ drawn from the conditional distribution $\Theta_{P|Z_n}$.

$$\mathcal{C}_P = \mathbb{E}_{Z_n} \mathbb{E}_{\Theta, \Theta' \sim \Theta_{P|Z_n}} \mathbb{E}_X[\mathrm{KL}(f(\Theta, X) \| f(\Theta', X))]$$

  Here, $\Theta_{P|Z_n}$ denotes the distribution over parameters resulting from applying procedure $P$ to dataset $Z_n$.

- **Local Sensitivity** ($S_\rho(w)$): Measures the expected maximum change in the model's output distribution within a $\rho$-radius ball around a specific parameter vector $w$. For consistency with the derivation below, we use the form where the expectation is inside the maximization.

$$S_\rho(w) = \max_{\|\delta\|_2 \leq \rho} \mathbb{E}_X[\mathrm{KL}(f(X, w + \delta) \| f(X, w))]$$

  Here, $\delta \in \mathbb{R}^d$ is a perturbation to the parameters $w$.

**Assumptions** The following derivation relies on several key assumptions:

1. **Isotropic Covariance Posterior Assumption**: For a given training set $Z_n$, the conditional parameter distribution $\Theta_{P|Z_n}$ can be approximated by an isotropic distribution centered at a specific parameter vector $w_{Z_n}$ derived from $Z_n$: $\mathbb{E}[\Theta_{P|Z_n}] = w_{Z_n}$, $\mathrm{Cov}[\Theta_{P|Z_n}] = s^2 \mathbf{I}_d$, where $s^2$ is a small variance. This approximation is motivated by studies interpreting

Stochastic Gradient Descent (SGD) as a form of approximate Bayesian inference, where the distribution of parameters after training can resemble a Gaussian centered near a mode of a posterior distribution related to the loss function [21].

2. **Validity of Second-Order KL Approximation**: The KL divergence between outputs of models with slightly different parameters can be accurately approximated by a quadratic form involving the Fisher Information Matrix (FIM). This relies on the parameter difference being small, implying $s^2$ must be small.

3. **Effective FIM Constancy in Expectation**: The variations of the FIM $F(\Theta')$ for $\Theta' \sim \mathcal{N}(w_{Z_n}, s^2 \mathbf{I}_d)$ around $F(w_{Z_n})$ are assumed to average out sufficiently within the expectation required to calculate $\mathcal{C}_{P|Z_n}$. This allows the approximation $\mathcal{C}_{P|Z_n} \approx s^2 \mathrm{Tr}(F(w_{Z_n}))$.

**Approximation of $\mathcal{C}_P$**  We first consider the conditional inconsistency for a fixed $Z_n$, denoted $\mathcal{C}_{P|Z_n}$, by removing the outer expectation $\mathbb{E}_{Z_n}$:

$$\mathcal{C}_{P|Z_n} = \mathbb{E}_{\Theta, \Theta' \sim \Theta_{P|Z_n}} \mathbb{E}_X[\mathrm{KL}(f(\Theta, X) \| f(\Theta', X))]$$

Applying the isotropic covariance posterior assumption, $\Theta = w_{Z_n} + \delta$ and $\Theta' = w_{Z_n} + \delta'$, where $\delta, \delta'$ are independent perturbations ($\mathbb{E}[\delta] = \mathbb{E}[\delta'] = 0, \mathrm{Cov}[\delta] = \mathrm{Cov}[\delta'] = s^2 \mathbf{I}_d$).

$$\mathcal{C}_{P|Z_n} \approx \mathbb{E}_{\delta, \delta'} \mathbb{E}_X[\mathrm{KL}(f(w_{Z_n} + \delta, X) \| f(w_{Z_n} + \delta', X))]$$

Using the second-order Taylor expansion for KL divergence taking the expectation over $X$, valid for small $\|\delta - \delta'\|$ (i.e., small $s^2$):

$$\mathbb{E}_X[\mathrm{KL}(f(w_{Z_n} + \delta, X) \| f(w_{Z_n} + \delta', X))] = \frac{1}{2}(\delta - \delta')^T F(w_{Z_n} + \delta')(\delta - \delta') + O(\|\delta\|^3)$$

Let $u = \Theta - \Theta' = \delta - \delta'$. Since $\delta, \delta'$ are independent, $u \sim \mathcal{N}(0, 2s^2 \mathbf{I}_d)$. Substituting this into the expression for $\mathcal{C}_{P|Z_n}$:

$$
\begin{aligned}
\mathcal{C}_{P|Z_n} &= \mathbb{E}_u\left[\frac{1}{2}u^T F(\Theta')u\right] + O(\|\delta\|^3) \\
&= \mathbb{E}_u\left[\frac{1}{2}u^T F(w_{Z_n})u\right] + O(\|\delta\|^3) \quad \text{(FIM Constancy in Expectation Assumption)} \\
&= \frac{1}{2}\mathrm{Tr}(\mathrm{Cov}(u)F(w_{Z_n})) + \frac{1}{2}\mathbb{E}[u]^T F(w_{Z_n})\mathbb{E}[u] + O(\|\delta\|^3) \\
&= \frac{1}{2}\mathrm{Tr}(2s^2\mathbf{I}_d F(w_{Z_n})) + 0 + O(\|\delta\|^3) \quad (\mathbb{E}[u] = 0) \\
&\approx s^2 \mathrm{Tr}(F(w_{Z_n}))
\end{aligned}
$$

Thus, the conditional inconsistency for a fixed $Z_n$ is approximately proportional to the trace of the FIM evaluated at $w_{Z_n}$:

$$\mathcal{C}_{P|Z_n} \approx s^2 \mathrm{Tr}(F(w_{Z_n})) \tag{15}$$

The overall inconsistency $\mathcal{C}_P$ is the expectation of this quantity over $Z_n$: $\mathcal{C}_P \approx \mathbb{E}_{Z_n}[s^2 \mathrm{Tr}(F(w_{Z_n}))]$.

**Approximation of $S_\rho(w_{Z_n})$**  Applying the same second-order KL approximation to the definition of $S_\rho(w_{Z_n})$:

$$S_\rho(w_{Z_n}) = \max_{\|\delta\|_2 \le \rho} \frac{1}{2}\delta^\top F(w_{Z_n})\delta + O(\|\delta\|^3)$$

The maximum value of the quadratic form $\delta^T A \delta$ for a positive semi-definite matrix $A$ subject to $\|\delta\|_2 \le \rho$ is achieved when $\delta$ is aligned with the eigenvector corresponding to the largest eigenvalue ($\lambda_{\max}(A)$) and has norm $\rho$. Thus:

$$S_\rho(w_{Z_n}) = \frac{1}{2}\rho^2 \lambda_{\max}(F(w_{Z_n})) \tag{16}$$

This shows that the local sensitivity $S_\rho$ is approximately proportional to the largest eigenvalue of the FIM.

**Connecting $\mathcal{C}_{P|Z_n}$ and $S_\rho(w_{Z_n})$** For a $d \times d$ positive semi-definite matrix $A$, the relationship between its trace and largest eigenvalue is given by $\frac{1}{d}\mathrm{Tr}(A) \leq \lambda_{\max}(A) \leq \mathrm{Tr}(A)$. Applying this to the FIM $F(w_{Z_n})$:

$$\frac{1}{d}\mathrm{Tr}(F(w_{Z_n})) \leq \lambda_{\max}(F(w_{Z_n})) \leq \mathrm{Tr}(F(w_{Z_n}))$$

Substituting this into the approximation for $S_\rho(w_{Z_n})$ from Eq. (16):

$$\frac{\rho^2}{2d}\mathrm{Tr}(F(w_{Z_n})) \leq S_\rho(w_{Z_n}) \leq \frac{\rho^2}{2}\mathrm{Tr}(F(w_{Z_n}))$$

Let's assume a plausible connection, for instance, $s^2 = \rho^2/d$. Substituting this into the approximation for $\mathcal{C}_{P|Z_n}$ from Eq. (15), we get $\mathcal{C}_{P|Z_n} \approx \frac{\rho^2}{d}\mathrm{Tr}(F(w_{Z_n}))$. Combining this with the bounds for $S_\rho(w_{Z_n})$:

$$\frac{1}{2}\left(\frac{\rho^2}{d}\mathrm{Tr}(F(w_{Z_n}))\right) \leq S_\rho(w_{Z_n}) \leq \frac{d}{2}\left(\frac{\rho^2}{d}\mathrm{Tr}(F(w_{Z_n}))\right)$$

This leads to the final approximate relationship between the conditional inconsistency (for a fixed $Z_n$) and the local sensitivity (at the corresponding $w_{Z_n}$):

$$\frac{1}{2}\mathcal{C}_{P|Z_n} \leq S_\rho(w_{Z_n}) \leq \frac{d}{2}\mathcal{C}_{P|Z_n} \tag{17}$$

This result suggests that, under the stated assumptions, the conditional inconsistency $\mathcal{C}_{P|Z_n}$ and the local sensitivity $S_\rho(w_{Z_n})$ are approximately proportional, with the proportionality factor potentially depending on the parameter dimension $d$.

**anisotropic covariance** Let $\mathrm{Cov}[\Theta_{P|Z_n}] = s^2\Sigma$, where $s^2 = \frac{\rho^2}{d}$. Starting from $\mathcal{C}_{P|Z_n} = \frac{1}{2}\mathrm{Tr}(\Sigma F(w_{Z_n}))$,

$$\lambda_{min}(\Sigma)\mathrm{Tr}(F) \leq \mathrm{Tr}(\Sigma F) \leq \lambda_{\max}(\Sigma)\mathrm{Tr}(F)$$

$$\lambda_{min}(\Sigma)\lambda_{\max}(F) \leq \mathrm{Tr}(\Sigma F) \leq \lambda_{\max}(\Sigma)d\lambda_{\max}(F)$$

$$\frac{\rho^2}{2d\lambda_{\max}(\Sigma)}\mathrm{Tr}(\Sigma F) \leq \frac{\rho^2}{2}\lambda_{\max}(F) \leq \frac{\rho^2}{2\lambda_{min}(\Sigma)}\mathrm{Tr}(\Sigma F)$$

$$\frac{1}{\lambda_{\max}(\Sigma)}\mathcal{C}_{P|Z_n} \leq S_\rho(w_{Z_n}) \leq \frac{d}{\lambda_{min}(\Sigma)}\mathcal{C}_{P|Z_n}$$

**Practical Considerations: Eigenvalue Spectrum of Neural Networks** In practice, for deep learning models, the FIM often exhibits a sparse eigenvalue spectrum: many eigenvalues are close to zero, and only a few are significantly large. In such cases:

- The trace $\mathrm{Tr}(F) = \sum \lambda_i$ is dominated by the sum of the few large eigenvalues.
- The ratio $\lambda_{\max}(F)/\mathrm{Tr}(F)$ might be closer to $1/m'$ than $1/d$, where $m' \ll d$ is the "effective rank" or number of dominant eigenvalues.

This implies that the bounds relating $\lambda_{\max}(F)$ and $\mathrm{Tr}(F)$ might be tighter than the general $1/d$ and $1$ factors suggest. Consequently, the relationship between $\mathcal{C}_{P|Z_n}$ (related to trace) and $S_\rho$ (related to max eigenvalue) could be closer to direct proportionality than Eq. (5) indicates, especially if $s^2$ is appropriately related to $\rho^2$.

**Summary and Limitations** This analysis provides a heuristic argument suggesting a connection between conditional inconsistency $\mathcal{C}_{P|Z_n}$ and local sensitivity $S_\rho(w_{Z_n})$. Under assumptions of a Gaussian posterior, small variance $s^2$, validity of second-order KL approximations, local FIM constancy, and a specific link between $s^2$ and $\rho^2$ (e.g., $s^2 = \rho^2/d$), we find that $S_\rho(w_{Z_n})$ is approximately proportional to $\mathcal{C}_{P|Z_n}$, potentially up to a factor related to dimension $d$. This connection is mediated by the trace and the maximum eigenvalue of the Fisher Information Matrix. The practical observation of sparse FIM eigenvalues might strengthen this relationship.

## C  Decision boundary of neural networks and principal eigenspace of FIM

To intuitively analysis the role of $\delta_1$ in training of neural network, we conducted experiments using 3-layer fully-connected neural network on two-dimensional synthetic data. the data is generated from a mixture of three Gaussian distributions, a setup analogous to that employed by [12] in their investigation of the characteristic of the FIM eigensubspace. Their work demonstrated that perturbing parameters along the principal eigenvectors of the FIM can lead to significant modifications in the decision boundary, such as increasing or decreasing the margins of specific classes.

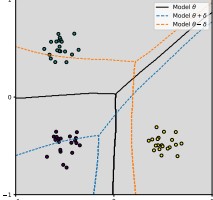 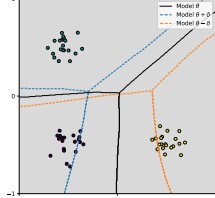 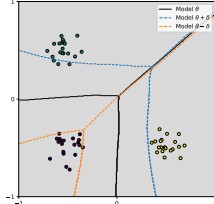 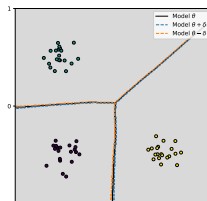

(a) Decision boundary perturbed by $\delta_1$ from $\varepsilon_1$  (b) Decision boundary perturbed by $\delta_1$ from $\varepsilon_2$  (c) Decision boundary perturbed by $\delta_1$ from $\varepsilon_3$  (d) Decision boundary perturbed by $\varepsilon$

Figure 4: A synthetic classification example. the black, blue, orange lines correspond to decision boundaries of the NN with trained parameter values, and parameter values perturbed by $\delta_1$. Each plot use different noise.

Our investigation focuses on whether $\delta_1$, despite being derived from only a single gradient step (as described in Algorithm 1) and thus influenced by an initial random noise vector $\varepsilon$, still induces substantial changes in the neural network's decision boundary. Figure 4 visualizes these effects. The black lines in each subfigures depict the original decision boundary obtained with the trained parameters $w$. Figure 4 (a-c) show the perturbed decision boundaries (blue and orange lines) when distinct $\pm\delta_1$ with $\rho = 0.5$ is added to $w$. Each of these $\delta_1$ vectors was computed using a different random initialization noise vector, denoted as $\varepsilon_1$, $\varepsilon_2$, and $\varepsilon_3$, respectively. For a direct comparison of the pertubation's nature, Figure 4(d) illustrates the decision boundary perturbed by directly adding the random noise vector $\varepsilon$ to $w$. This vector $\varepsilon$ is sampled from same distribution as initial vectors (e.g.$\varepsilon_1$) and, is scaled to $\|\varepsilon\|_2 = \rho$ same with $\delta_1$. As observed in Figure 4 (d), direct perturbation with such an arbitrary random noise vector does not meaning fully alter the decision boundary, even when its norm is equivalent to that of the $\delta_1$. This is sharply opposed with the significant changes induced by $\delta_1$ perturbations shown in Figures 4 (a-c), underscoring that the direction derived by Algorithm 1, even in a single step, is substantially more influential than arbitrary noise of the same magnitude. This result intuitively suggest that the perturbation $\delta_1$ with single gradient step still meaningful and aligning with principle eigen vectors of FIM.

To investigate the alignment between the single-step perturbation vector $\delta_1$ and principle eigenspace of FIM, we explicitly calculate the FIM and its top three eigenvector $v_1$, $v_2$, and $v_3$, corresponding to largesst eigenvalues $\lambda_1 > \lambda_2 > \lambda_3$. The perturbation $\delta_1$, results from one normalized gradient ascent step applied to the KL divergence objective, starting from an initial random noise $\varepsilon$. In terms of power iteration algorithm, the $\delta_1$ after first iteration without normalization, is sum of eigenvector of FIM weighted by $\lambda_i \alpha_i$.

Formally, let the initial random noise $\varepsilon$ be expressed in the eigenbasis of $F(w)$ as $\varepsilon = \sum_i^m \alpha_i v_i$. $\varepsilon \sim \mathcal{N}(0, \sigma^2 I_m)$, then the coefficient $\alpha_i$ are i.i.d. as $\mathcal{N}(0, \sigma^2)$ since $\{v_i\}$form an orthonormal basis.

$$F(\theta)\varepsilon = \sum_i^m \lambda_i v_i v_i^\top \sum_i^m \alpha_i v_i$$

$$= \sum_i^m \lambda_i \alpha_i v_i$$

So cosine similarity between $\delta_1$ and $v_i$ is $\lambda_i\alpha_i$. And $\frac{\|\sum_i^3 \delta_1^T v_i\|}{\|\delta_1\|}$, which indicates how much the $\delta_1$ is in principle eigen space, $\{u|u = av_1 + bv_2 + cv_3, \quad abc \in [0,1]\}$ of FIM, is $\frac{\|\sum_i^3 \alpha_i\lambda_i v_i\|}{\|\delta\|}$.

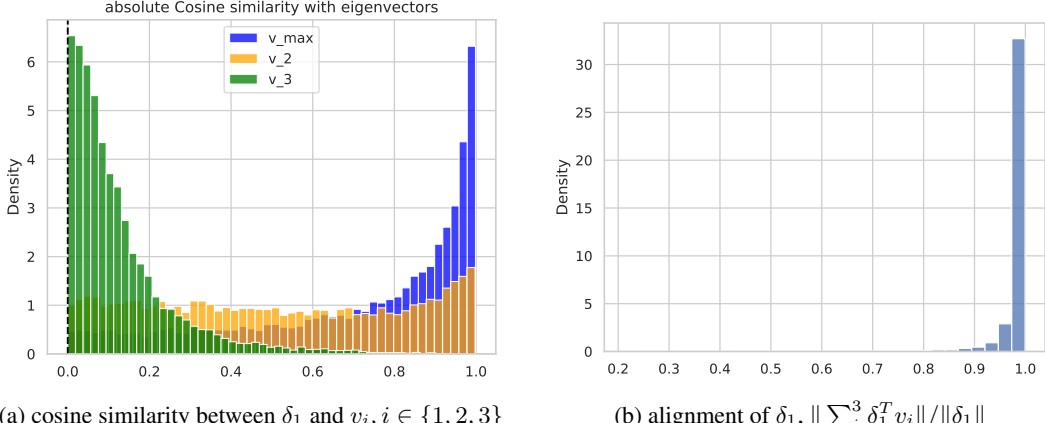

(a) cosine similarity between $\delta_1$ and $v_i, i \in \{1, 2, 3\}$     (b) alignment of $\delta_1$, $\|\sum_i^3 \delta_1^T v_i\|/\|\delta_1\|$

Figure 5: A synthetic classification example. $\delta_1$ are align with top three eigen Vector of FIM sampling from 10000 gaussian noises $\varepsilon$

Figure 5 presents empirical results from this analysis. Figure 5 (a) shows histograms of the absolute cosine similarities between $\delta_1$ (generated from 10,000 different $\varepsilon$ samples) and each of the top three eigenvectors $v_1$, $v_2$, and $v_3$. We observe that $\delta_1$ tends to have a higher cosine similarity with $v_1$ (corresponding to the largest eigenvalue $\lambda_1$) compared to $v_2$, and $v_3$. Furthermore, Figure 5 (b) displays the distribution of the squared norm of the projection of $\delta_1$ onto the top-3 eigenspace. The values are predominantly close to 1, indicating that $\delta$ vectors derived from different initial noise samples are largely confined to this principal subspace. These results empirically support the theoretical expectation that the single-step perturbation $\delta_1$ is predominantly aligned with the principal eigenspace of the FIM.

# D   IAM algorithms

---

**Algorithm 2** Inconsistency-Aware Minimization (SAM-like variant: IAM-S)

---

1: **Input:** Initial model parameters $\theta^0$; Learning rate $\eta$; neighborhood size $\rho$; training set $Z_n$; Batch size $b$; Number of steps $K$ for Algorithm 1.
2: **while** not converged **do**
3:     Sample batch $\{(x_i, y_i)\}_{i=1}^b$.
4:     Compute $\delta_K$ from Algorithm 1 using current $\theta$, $\rho$, and data $\{x_i\}_{i=1}^b$.
5:     Compute gradient $g = \nabla_\theta L(\theta)|_{\theta+\delta_K}$
6:     Update parameters: $\theta \leftarrow \theta - \eta g$.
7: **end while**
8: **Return** optimized parameters $\theta$.

---

# E   Experimental Details

**Practical Considerations in estimating $S_\rho(\theta)$**

- **Computational Efficiency:** Calculating the FIM explicitly and performing eigenvalue decomposition is computationally expensive ($O(m^2)$ or worse, where $m$ is the number of parameters). Algorithm 1 avoids this by requiring only $K$ gradient computations (forward and backward passes) per estimation, making its computational cost approximately $O(mK)$, which is significantly more feasible for large networks.

- **Number of Steps (K):** Empirical studies on neural network Hessians and FIMs suggest that the eigenspectrum is often dominated by a huge largest eigenvalues. Thus, the Power Iteration method can converge quickly to the dominant eigenvector. In practice, using a small number of steps, often just $K = 3$, is found to be sufficient to get a reasonable estimate of the maximizing direction. This makes the computation highly efficient.

- **Averaging for reduce Variance from initialization:** The estimate of $S_\rho(w)$ obtained from Algorithm 1 depends on the random initialization $\delta_0$ with just $K = 1$. To obtain a more stable estimate, we compute the metric multiple times (e.g., 10 times) with different random initializations for $\delta_0$ and report the average value: $\mathbb{E}_{\delta_0}[\text{Estimate from Alg 1}]$.

**Infrastructure** Experiments are implemented in PyTorch2.7 and executed on NVIDIAA40 and L4 GPUs.

### E.1 Image classification

Each reported metric is the mean$\pm$ standard error computed over minimum test error from three independent runs.

**Dataset.** We evaluate on the **CIFAR-10** (50,000 training, 10,000 test images) and **CIFAR-100** (50,000 training, 10,000 test images). All images are resized to $32 \times 32$ and preprocessed with

- *RandomCrop*(32, padding= 4),
- *RandomHorizontalFlip*($p = 0.5$), and
- *Normalization* using the official mean and standard deviation.

No additional augmentation such as Cutout or Mixup is applied.

**Optimization.** Models are trained for **200 epochs** with mini-batch size **128**. We use SGD with momentum 0.9, weight decay $5 \times 10^{-4}$ as an optimizer, and a multistep learning rate schedule that decays the initial rate 0.1 by 0.2 at epochs 60, 120, and 160.

**Hyperparameters.** The inconsistency weight $\beta$ and neighborhood radius $\rho$ are selected from $\beta \in \{0.1, 1.0, 5.0, 10.0, 20.0\}$ and $\rho \in \{0.01, 0.05, 0.1, 0.5, 1.0\}$ via grid search on the validation split using 10% of the training dataset. The best pairs are $(1.0, 0.1)$ for CIFAR-10 and $(10.0, 0.1)$ for CIFAR-100. For IAM-S, 0.1 and 0.5 were selected $\rho$ value for CIFAR-10, 100 respectively.

**Loss function.** Cross-entropy with label smoothing ($\alpha = 0.1$) is used for all methods.

### E.2 Semi-supervised learning

In semi-supervised learning experiment, we shared most of the settings with image classification. Each reported metric computed over minimum test error from three independent runs.

**Optimization.** Models are trained for **100 epochs** without learning rate scheduling.

**Hyperparameters.** We used $\beta = 1.0$ and $\rho = 0.1$ for both CIFAR-10 and CIFAR-100. SAM is also trained with $\rho = 0.1$.

### E.3 Self-supervised learning

Each reported metric is the mean **test accuracy** obtained from three independent runs.

**Dataset.** We use the **CIFAR-10** benchmark. All images are resized to $32 \times 32$ and augmented with the SimCLR[6] pipeline:

- *RandomResizedCrop*(32, scale=$(0.4, 1.0)$),
- *RandomHorizontalFlip*($p = 0.5$),

- *ColorJitter*$(0.4, 0.4, 0.2, 0.1)$ with probability $0.8$,
- *RandomGrayscale*$(p{=}0.2)$, and
- *Normalization* using the official mean and standard deviation.

**Encoder&Projection Head.**    We adopt a **ResNet-18** backbone with the first convolution modified to $3\times3$ layer with stride $= 1$ and the max-pool removed. The projector is a two-layer MLP (hidden size $512$, output size$128$) with ReLU activation.

**Optimization.**    Models are trained for **200 epochs** with mini-batch size **1024**. We use SGD (momentum $0.9$, weight decay $1\times10^{-4}$) and a cosine-annealing learning-rate schedule starting at $1.0$ after a 10-epoch warm-up.

**Contrastive Loss.**    The NT-Xent loss is computed with temperature $\tau{=}0.5$.

**IAM Hyperparameters.**    We set the inconsistency weight $\beta{=}1.0$, neighborhood radius $\rho{=}0.1$, and noise-scale $3.0$ (Gaussian initialization). The local inconsistency is computed between projection head outputs with temperature $\tau{=}0.5$.

**Stability Heuristics.**    Is identical to image classification setting.

**Linear Evaluation.**    After every 5 epochs (and at the final epoch), a frozen encoder is evaluated via a linear probe trained for 20 epochs with AdamW optimizer on the full training set (batch size 1024). The reported metric is the probe's test accuracy.

