# OpenReview forum: "Inconsistency-Aware Minimization: Improving Generalization with Unlabeled Data"
_NeurIPS.cc/2025/Conference — Submitted to NeurIPS 2025_

### Official Review · Reviewer_aXXR · 2025-06-29

**Clarity:** 3
**Significance:** 3
**Originality:** 3
**Rating:** 5
**Confidence:** 3

**Summary:**

This paper proposed inconsistency-aware minimization, a new method that restricts the change of the neural network model in the output function space to improve the generalization ability. The method uses the KL divergence under a perturbation to measure the local inconsistency, which can be estimated using unlabeled data in practice. Experimental results on CIFAR 10 and CIFAR 100 confirm that the proposed method can improve the generalization performance over existing methods.

**Questions:**

1. The IAM algorithm seems to be computationally more expensive than SAM in [1], it is good to include the computation comparison, given that the performance improvement is rather marginal. In particular, it seems that Algorithm 1 in [1] also requires one trained model only. Can you explain more about the claim in lines 137-138?

2. Is there a reason why KL divergence is preferred to measure the inconsistency instead of other metrics, such as L2 distance?

3. How many benefits can we gain from increasing the number of iterations K in Algorithm 1? It would be nice to provide some experimental evidence on this.

[1] P. Foret, A. Kleiner, H. Mobahi, and B. Neyshabur. Sharpness-aware minimization for efficiently
374 improving generalization, 2021

**Ethical Concerns:**

["NO or VERY MINOR ethics concerns only"]

**Final Justification:**

The authors have resolved my concern. Overall, I have no outstanding concern about this paper, but I acknowledge the opinion from another reviewer that the novelty of the methodology is limited. I would like to keep my score.

**Limitations:**

Yes

**Paper Formatting Concerns:**

No concern

**Quality:**

3

**Strengths And Weaknesses:**

Strengths: The proposed metric of local inconsistency and the IAM method provide an interesting perspective on the generalization of neural networks. The core strengths are:
1. The new IAM methods show empirical improvements upon classical SAM methods across various settings. Especially, the author shows the subgroup behaviour in Figure 1, which is nice to understand why local inconsistency could be a promising replacement to sharpness.

2. The proposed method can be computed using the cheap unlabeled data, making it a promising approach to measure and improve generalization in practice.

Weakness: Overall, the improvement upon existing works needs to be further justified.
1. A similar inconsistency measure has already been studied previously in [1], and it is unclear whether the proposed metric has advantages in practice. It would be good to make an empirical comparison to this metric in addition to the current high-level justification.

2. The authors have mentioned several drawbacks of using sharpness as the metric for generalization. While Figure 1 is a nice illustration for showing the difference, it would be good to provide some intuition to understand what makes the two metrics different in a geometric perspective, and to what extent it can solve the drawbacks of sharpness in the literature.

[1] R. Johnson and T. Zhang. Inconsistency, instability, and generalization gap of deep neural
395 network training, 2023. URL https://arxiv.org/abs/2306.00169

---

> ### Author Rebuttal · Authors · 2025-07-30
>
> - Inconsistency is a statstical measure while local inconsistency is a **optimizable generalization measure**.
> - IAM **doesn't require additional backward pass**, but one more forward pass compared to SAM.
>
> ---
> >**W1** A similar inconsistency measure has already been studied previously in [1], and it is unclear whether the proposed metric has advantages in practice. It would be good to make an empirical comparison to this metric in addition to the current high-level justification.
>
> Our proposed "local inconsistency" and the Inconsistency-Aware Minimization (IAM) framework are **fundamentally novel in their (i) practical computability (sigle model, few gradient steps), and (ii) ultimate objective (acts as regularizer).**
> The term "inconsistency" is used by both papers, but it refers to two different concepts.
>
> - **Inconsistency in Johnson & Zhang (J&Z)**: This is a **procedural, statistical measure** of a stochastic **training procedure**, denoted as $P$. It is defined as $$\mathcal{C}\_{P}=\mathbb{E}\_{Z_n,Θ,Θ'\sim Θ_{P|Z_{n}}}\mathbb{E}_{x}KL(f(x, Θ)||f(x, Θ')).$$ This metric quantifies the expected output disagreement between *different models* ($θ$ and $θ'$) that result from the *same stochastic training process*. It is a property of the model-generating distribution, not of a single model.
>
> - **Our Local Inconsistency**: This is a **local, geometric property** of a **single, specific model parameter vector**, $θ$. It is defined as $$S_{\rho}({\color{blue}θ})={\color{red}\max_{||δ||\le\rho}}\mathbb{E}_{x}[KL(f(x,{\color{blue}θ})||f(x,{\color{blue}θ}+δ))].$$ This metric quantifies the **worst-case** sensitivity of a *specific model's output* to perturbations in its parameter space. It measures the local geometry of a given solution.
>
> In essence, J&Z's inconsistency analyzes the **randomness of the training process**, while our local inconsistency analyzes the **local geometry of the solution**.
>
> #### 2. Innovation in Practical Computability
> This conceptual difference leads to a critical innovation in practicality and computational feasibility, as decribed in line 140-143 on the paper.
> * **J&Z's Inconsistency** is an **analytical tool**. To estimate it, one must train multiple models (e.g., 8) on multiple disjoint training sets (e.g., 4 sets), making it computationally prohibitive to use as a direct regularizer within a single training run.
> * **Our Local Inconsistency** is designed to be a **optimizable measure**. It can be efficiently approximated for a **single model** using only a few gradient steps (often K=1). This single-model computability is the key that unlocks its use as a **regularizer**.
>
> #### 3. Algorithmic Novelty: From Analysis to Optimization
>
> The most significant novelty of our work is the transformation of an analytical concept into a **practical** optimization framework.
>
> Due to computational demands, "inconsistency" can't be directly minimized during single model training.
>
> Our work bridges this critical gap. By defining a computable, single-model metric, we introduce the **IAM** framework. IAM-D and IAM-S are novel algorithms that actively seek solutions in regions of low output sensitivity.
>
> ---
> >**W2** The authors have mentioned several drawbacks of using sharpness as the metric for generalization. While Figure 1 is a nice illustration for showing the difference, it would be good to provide some intuition to understand what makes the two metrics different in a geometric perspective, and to what extent it can solve the drawbacks of sharpness in the literature.
>
> The fundamental geometric difference between **sharpness** and **local inconsistency** lies in what they measure and, critically, what data they use for that measurement. While both assess the geometry around a solution in the parameter space, sharpness evaluates the curvature of the **loss landscape** defined by the **training data**, whereas local inconsistency measures the sensitivity of the model's output distribution using **unlabeled, held-out data**. This distinction is key to understanding why local inconsistency can overcome some of the documented drawbacks of sharpness.
>
> ### Geometric view of Sharpness
> From a geometric standpoint, sharpness quantifies the local curvature of the training loss surface at a given parameter vector, $\theta$. It is commonly measured by the trace or the maximum eigenvalue of the loss Hessian matrix. A high sharpness value indicates a steep and narrow minimum, where small perturbations to the model's parameters cause a large increase in the training loss.
>
> The primary drawback, as illustrated in the document, is that this geometric view is entirely contingent on the training data.
>
> ### Geometric view of local inconsistency
>
> In contrast, local inconsistency, $S_ρ(θ)$, offers a different geometric perspective. It is defined as the maximum expected KL divergence between the model's output distributions.
> $f(x,θ)$ and $f(x,θ+δ)$, under a parameter perturbation $δ$.
>
> Crucially, this metric is computed using a disjoint, unlabeled dataset. Geometrically, this means local inconsistency is not measuring the curvature of the loss function, but rather the sensitivity of the function that the model itself computes.
>
> ---
> >**Q1** The IAM algorithm seems to be computationally more expensive than SAM in [1], it is good to include the computation comparison, given that the performance improvement is rather marginal. In particular, it seems that Algorithm 1 in [1] also requires one trained model only. Can you explain more about the claim in lines 137-138?
>
> - Computational expensivity is totally not large compared to performance gain.
> - Local inconsistency can be calulated in single model unlike “inconsistency”, and can be estimated using unlabeled data unlike sharpness-based measures.
> #### 1. Concerns about computational burden
> IAM requires three forward passes and two backward pass. Therefore, total computational burden is approximately 2.5 times that of SGD (SAM is 2.0x SGD). We attach a time anlaysis of a forward and backward pass for each methods, average of 100 iterations.
>
> |Optimizer|# of Forward operations |# of backward operation | forward/backward time (cifar 10) |Total Cost (vs. SGD)| runnig time on ImageNet |
> |-|-|-|-|-|-|
> |SGD| 1| 1|31.33, 49.41 ms/step|  1.0x|533  s/epoch|
> |SAM| 2| 2|51.26, 95.48 ms/step| ~2.0x|-|
> |IAM| 3| 2|83.63, 99.10 ms/step| ~2.5x|1051 s/epoch|
>
> On top of that, there are additional empirical investigations that prove IAM’s ability to generalize better. It is comparable or even surpass ASAM’s performance.
>
> Table: Test Error (mean ± stderr) of different optimization methods on various datasets
> |Dataset|SGD|SAM|ASAM|IAM-D|IAM-S|
> |-|-|-|-|-|-|
> |CIFAR-10 |3.95 ± 0.05 | 3.31 ± 0.01 |**3.15** ± 0.02|3.28 ± 0.06|3.28 ± 0.03|
> |CIFAR-100|19.17 ± 0.19| 17.63 ± 0.12|17.15 ± 0.11|17.16 ± 0.03|**16.82** ± 0.01|
> |F-MNIST|4.45 ± 0.05|4.13 ± 0.02|4.11 ±<0.01|4.13 ± 0.04|**4.10** ± 0.05|
> |SVHN|3.82 ± 0.06|3.47 ± 0.09|3.24 ± 0.04|**3.13** ± 0.06|**3.13.** ± 0.01|
>
> #### 2. Additional explanation about single-model computability.
> “Inconsistency” is a powerful measure, but it is computed over different models. Sharpness based measures are based on loss funciton, which requires explicit labels. On the other hand, local inconsistency enjoys the ability to be computed on a single model, while keeping the label-agnostic nature of “inconsistency”.
>
> ---
> >**Q2** Is there a reason why KL divergence is preferred to measure the inconsistency instead of other metrics, such as L2 distance?
>
> Kullback-Leibler (KL) divergence is favored over other metrics, such as the L2 distance, for measuring inconsistency due to its fundamental connection to information theory and the geometry of the model's parameter space.
>
> While KL divergence is a more general measure for the "distance" between probability distributions, it relates to the L2 distance under specific assumptions. For instance, if a model's output is assumed to be the mean of a Gaussian distribution with constant variance (i.e. $p(x;\theta) =\mathcal{N} (NN_{\theta}, \sigma^2I)$), the KL divergence simplifies to a function of the L2 norm between the means of the original and perturbed outputs:
>  $$ KL(p(x;\theta)||p(x;\theta+\delta)) = c||NN_{\theta}-NN_{\theta+\delta}||^2 + C.$$
>
> In this specific case, minimizing the KL divergence is equivalent to minimizing the squared L2 distance between the model's output vectors. However, KL divergence provides a more principled and general approach that is applicable to any probabilistic output, not just those that fit a simple Gaussian model, and it maintains a direct link to the underlying information geometry of the learning problem.
>
> ---
> >**Q3** How many benefits can we gain from increasing the number of iterations K in Algorithm 1? It would be nice to provide some experimental evidence on this.
>
> Mentioned in line 249-250, our experiments were conducted with **K=1** for Algorithm 1. Empirically local inconsistency almost converges when $K=3$ (see figure in markdown file or figures folder in supplemental metrials). Furthermore, estimated $S_\rho(\theta)$ values with both $K=3$ and $K=1$ are highly correlated with Pearson correlation coefficient, being above 0.9 in trained models for figure 1.

---

> > ### Comment · Reviewer_aXXR · 2025-08-03
> >
> > Thank you for the responses. My questions has been properly addressed.

---

### Official Review · Reviewer_8R8r · 2025-07-01

**Clarity:** 3
**Significance:** 2
**Originality:** 2
**Rating:** 3
**Confidence:** 4

**Summary:**

- Introduce a novel generalization metric called local inconsistency, defined via small perturbations of the model parameters $\theta$.
Theoretically, connect local inconsistency to the largest eigenvalue of the Fisher information matrix.
- The idea of estimating inconsistency from unlabeled data involves multiple models on the same dataset with different random seeds, proposed by Zhang et al. [15], with extensive theoretical analysis and experiments.
- Propose the Inconsistency-Aware Minimization (IAM) framework (with variants IAM-D and IAM-S) to directly reduce local inconsistency, which is an alternative method to measure generalization compared to the sharpness measure used in SAM.
- Evaluate on the CIFAR dataset, comparing IAM with SGD and SAM in various label settings.

**Questions:**

- The authors state IAM-D uses CE on the labeled subset and local inconsistency on both the labeled and unlabeled subsets. What is the exact setting of this experiment (in terms of how these losses are balanced, the schedules for applying each term, etc)?
- In Self-Supervised Learning (SSL), SGD is the common optimizer to evaluate and is a reasonable baseline. However, modern self-supervised methods often benefit from optimizers like AdamW or SogCLR-style, which often yield stronger generalization.
- Experiments are limited to CIFAR variants; including larger benchmarks (ImageNet, Food101, etc.) would better demonstrate scalability.
- After SAM, many approaches have been proposed to improve SAM, and also been pointed out that the sharpness measurement in SAM is not reliable. It would be more appropriate to compare IAM against other works such as ASAM, GSAM, LookSAM, fSAM, etc
-  Key methods such as FisherSAM—which measures parameter perturbation via the Fisher information—are absent from comparisons and discussion, despite their conceptual overlap with the  $\delta$ term.
- The author discussed in section  5.4 that IAM implicitly minimizes output entropy and acts like a regularization, but no experiments currently validate this effect, and compared to the mentioned methods (LR, ER).
- Odd choice of hyperparameters for $K, \beta$, and $\rho$, which are based on grid search and only give the value for the best performance without a sensitivity analysis.

**Ethical Concerns:**

["NO or VERY MINOR ethics concerns only"]

**Final Justification:**

**1. Comparison to TRADES**
There is no difference in the optimization problem between TRADES and IAM-D except for the perturbation space. (In TRADES, the data $x$ can be treated as a parameter when finding the perturbation, starting from the benign data as the initial value. The perturbation $\delta$ is also randomly initialized before being updated.)
 All three methods — TRADES, IAM-D, and IAM-S — maximize the KL term to find the perturbation $\delta$. The key differences are in the model update step:

TRADES:  minimizes $L\_{\text{TRADES}}(\mathbf{\theta}) = \frac{1}{n} \sum\_{i=1}^{n} \left( \text{CE}(f(\mathbf{x}\_i, \theta), y\_i) + \beta \cdot \max_{||\delta|| \leq \rho} \text{KL}(f(\mathbf{x}\_i, \theta) \parallel f(\mathbf{x + \delta}, \theta) ) \right)$

IAM-D minimizes $L\_{\text{IAM-D}}(\mathbf{\theta}) = \frac{1}{n} \sum\_{i=1}^{n} \left( \text{CE}(f(\mathbf{x}\_i, \theta), y\_i) + \beta \cdot \max_{||\delta|| \leq \rho} \text{KL}(f(\mathbf{x}\_i, \theta) \parallel f(\mathbf{x}, \theta + \delta) ) \right)$

IAM-S minimizes $L\_{\text{IAM-S}}(\mathbf{\theta}) = \frac{1}{n} \sum\_{i=1}^{n} \text{CE}(f(\mathbf{x}\_i, \theta + \delta), y\_i) $

This makes IAM appear as a combination of SAM, TRADES, and prior works from J & Z.

---

**2. Contribution**

**Strengths**

1. IAM is flexible and can be applied to both supervised learning (SL) and semi-supervised learning (SSL).
2. The idea of minimizing *local inconsistency* alongside sharpness is interesting and potentially impactful.

**Concerns**

**SSL Setting:** While IAM shows promise for semi-supervised and contrastive learning (with few or no labels), the authors only compare IAM to SAM and SGD, which are not particularly strong baselines for SSL. In self-supervised learning, SGD is a reasonable starting point, but modern methods often use optimizers like AdamW, SogCLR-style approaches, which typically yield stronger generalization. Without comparisons to such optimizers, the strength of IAM in SSL remains unclear.

**SL Setting:** The experiments show IAM surpasses SAM and SGD in SL, but training takes significantly longer. While the approach of minimizing local inconsistency alongside sharpness is promising, the experimental setup lacks the rigor needed to make IAM stand out as a superior method.

For these reasons, I will keep my score.

**Limitations:**

Yes

**Quality:**

2

**Strengths And Weaknesses:**

## Strength:
- Introduce new generalization measurement, termed local inconsistency, based on model theta and its perturbation.
- Wide range of experiments from full labelled, limited sample, and no labelled sample.
- The paper is easy to follow.

## Weakness:
- The novelty of IAM in the fully supervised setting is limited: the “inconsistency” term was already introduced and analyzed in Zhang et al. [15], with extensive theoretical analysis and experiments.
- The authors proposed the $S_p$ that closely mirrors FisherSAM [1] criterion—both measure small-perturbation effects via KL divergence in output space. Even though the perspective leading to this term is not exactly the same, it should be discussed and compared in this work.
- The update form of IAM-D in formula 6, which perturbs model parameters, is similar to Virtual Adversarial Training (VAT) [2], which perturbs in the input space.
- While IAM shows promise for semi-supervised and contrastive learning (with few or no labels), the paper fails to discuss or compare its methods against existing work in those areas. Authors only compare IAM to SAM and SGD, which are not strong competitors in those areas.
- There is no discussion and analysis on the theoretical superiority of IAM (convergence rate and training time compared to other optimization methods)

[1] Kim, Minyoung, et al. "Fisher sam: Information geometry and sharpness aware minimisation." International Conference on Machine Learning. PMLR, 2022.
[2] Miyato, Takeru, et al. "Virtual adversarial training: a regularization method for supervised and semi-supervised learning." IEEE transactions on pattern analysis and machine intelligence 41.8 (2018): 1979-1993.
## Suggestion:
- Apply this method in case of using extra unlabelled samples for training, in which the labelled dataset is ImageNet and the extra dataset is unlabelled (could be CIFAR or Food101, etc) to enrich the training process
- Analyze this method from other perspectives, such as noisy label or out-of-distribution, which should be more reasonable

---

> ### Author Rebuttal · Authors · 2025-07-30
>
> >**W1.** Regarding the novelty of IAM in fully supervised setting compared to “inconsistency”
>
> Our proposed "local inconsistency" and Inconsistency-Aware Minimization (IAM) framework are **fundamentally novel in their (i) conceptual definition (locality, worst-case sensitivity), (ii) practical computability (sigle model, few gradient steps), and (iii) ultimate objective (acts as regularizer).**
>
> #### 1. Fundamental Conceptual Distinction
> The term "inconsistency" is used in both papers, but refers to two different concepts.
>
> - **Inconsistency in Johnson & Zhang (J&Z)**: This is a **procedural, statistical measure** of a stochastic **training procedure**. It is defined as $$\mathcal{C}\_{P}=\mathbb{E}\_{Z\_n,Θ,Θ'\sim Θ\_{P|Z\_{n}}}\mathbb{E}\_{x}KL(f(x, Θ)||f(x, Θ')).$$ This metric quantifies the expected output disagreement between *different models* ($Θ$ and $Θ'$) that result from the *same stochastic training process*. It is a property of the model-generating distribution, not of a single model.
>
> - **Our Local Inconsistency**: This is a **local, geometric property** of a **single, specific model parameter vector**, $θ$. It is defined as $$S_{ρ}({\color{blue}θ})={\color{red}\max_{||δ||\le ρ}}\mathbb{E}_{x}[KL(f(x,{\color{blue}θ})||f(x,{\color{blue}θ}+δ))].$$ This metric quantifies the **worst-case** sensitivity of a *specific model's output* to a perturbation in its parameter space. It measures a local geometry in a given solution.
>
> In essence, J&Z's inconsistency analyzes **randomness of the training process**, while our local inconsistency analyzes **local geometry of the solution**.
>
> #### 2. Innovation in Practical Computability
> This conceptual difference leads to a critical innovation in computational feasibility, as decribed in paper line 140-143.
> * **J&Z's Inconsistency** is an **analytical tool**. To estimate it, one must train multiple models (e.g., 8) on multiple disjoint training sets (e.g., 4 sets), making it computationally prohibitive as a direct regularizer.
> * **Local Inconsistency** is designed to be a **optimizable measure**. It can be efficiently approximated for a **single model** using only one or few gradient steps. This single-model computability is the key that unlocks usage as a **regularizer**.
>
> #### 3. Algorithmic Novelty: From Analysis to Optimization
>
> The most significant novelty of our work is transformation of an analytical concept into a **practical** optimization framework.
>
> Due to computational demands, "inconsistency" can't be directly minimized during single model training.
>
> Our work bridges this critical gap. By defining a computable, single-model metric, we introduce the **IAM** framework. IAM-D, -S are novel algorithms that actively seek solutions in low output sensitivity regions.
>
> ---
> > **W2. Q5.** Regrading concern about absent of comparisons and discussion with FisherSAM
> #### 1. Difference in Maximization Objective
>
> The most critical distinction lies in *what* is being maximized within the local neighborhood.
>
> * **FisherSAM** aims to find a solution robust to the **worst-case loss**. The objective is $\max_{ϵ^{\top}F(θ)ϵ \le γ^{2}}\mathbb{E}_{x, y} [CE(f(x,θ+ϵ), y)]$, which requires **labeled data**.
>
> * **IAM**, in contrast, seeks for a solution robust to **worst-case output inconsistency**. It directly maximizes the KL divergence between the model's outputs before and after perturbation:
> $\max_{||δ|| \le ρ} \mathbb{E}_{x}[KL(f(x,θ)||f(x,θ+δ))]$.
> The goal is to find regions in the parameter space where model has stable predictive distribution. This makes our measure **label-agnostic** and directly applicable to **unsupervised** or **semi-supervised** settings.
>
> ---
> #### 2. Difference in Neighborhood Definition and Perturbation Direction
>
> Differing objectives naturally lead to different definitions of the "neighborhood" and an adversarial perturbation directions.
>
> - **FisherSAM** defines neighborhood as an **ellipsoid** shaped by the FIM, $F(θ)$. The resulting perturbation is $F(θ)^{-1}\nabla l(θ)$. To caculate inverse of FIM, FisherSAM approximates FIM only with digonal term.
>
> - **IAM** defines neighborhood as a standard **Euclidean ball**. Because the objective is KL divergence itself, the gradient ascent is taken with respect to the KL divergence. Using the second-order approximation $KL(\dots) \approx \frac{1}{2}δ^T F(θ) δ$, the gradient direction is $F(θ)δ$. This direction maximizes *output divergence*, which is fundamentally different from those of SAM or FisherSAM.
>
> |Feature|SAM|FisherSAM|IAM (Ours)|
> |-|-|-|-|
> |Maximization Objective| Loss: $l(θ+ϵ)$ |Loss: $l(θ+ϵ)$|**Output Divergence**: $KL(f(θ)\|\|f(θ+δ))$|
> |Neighborhood Constraint| Euclidean Ball: $\|ϵ\|_2 \le γ$ | **FIM Ellipsoid**: $ϵ^{\top}F(θ)ϵ \le γ^{2}$ | Euclidean Ball: $\|δ\|_2\le ρ$ |
> |Perturbation Direction|$\nabla l(θ)$|$\hat{F}^{-1}\nabla l(θ)$|$F(θ)\varepsilon$ (K=1)|
> |Label Requirement|Required|Required|**Not Required**|
>
> As the table illustrates, IAM is not a variation of FisherSAM. Introducing a novel objective directly targeting stability of model's output distribution, IAM can be computed without labels and results in a unique perturbation direction distinct from both SAM's gradient ascent and FisherSAM's preconditioned ascent.
>
> ---
> >**W3.** Regarding seeming similarity to VAT
>
> We clarify that they are different in their objective.
>
> The core distinction lies in what is being perturbed:
> - VAT perturbs **input data** to find an adversarial direction in the input space, to smooth model's output with respect to local changes in the **data manifold**.
> - IAM perturbs **model parameters** to find an adversarial direction in the parameter space, to find solutions in "flat" regions of the **parameter landscape** where the model's function is stable.
>
> ---
> >**W4.** Regarding concerns about comparison with strong competitors in semi-supervised learning
>
> IAM can be readily applied to SSL scenarios by regularizing local inconsistency of unlabeled data. This allows leveraging second-order information from abundant unlabeled samples.
>
> We have validated this by integrating IAM-D into **FixMatch**, a powerful and widely-used SSL method. Simply adding local inconsistency term as an additional regularizer to original FixMatch objective is all.
>
> The results show that integration of IAM-D significantly improves the performance of FixMatch.
>
> Table: Test Error (mean ± stderr) on CIFAR-10 with 250 labels using WRN-28-2 model.
> |Method|Test Error (%)|
> |-|-|
> |FixMatch|6.26 ± 0.39|
> |FixMatch + IAM-D|**5.30** ± 0.08|
>
> This result demonstrates that IAM can serve as an effective "plug-and-play" regularizer enhancing existing SOTA SSL frameworks.
>
> ---
> >**W5.** Theoretical superiority of IAM with convergence rate.
>
> Convergence rate is **not our main focus**. IAM is focused on performance enhancement and generalization.
>
> Time analysis below shows time analysis compared to other optimization methods.
> |Optimizer|# of forward operations|# of backward operations|cifar 10 forward/backward time (ms/step)|total cost (vs. SGD)|runnig time on ImageNet|
> |-|-|-|-|-|-|
> |SGD| 1| 1|31/49|  1.0x|533  s/epoch|
> |SAM| 2| 2|51/95| ~2.0x|-|
> |IAM| 3| 2|84/99| ~2.5x|1051 s/epoch|
>
> ---
> >**Q1.** Settings of custom semi-supervised experiment.
>
> To show it's applicability, we calculated $S_ρ(\theta)$ with same size of (un)labeled data and add it to CE on labeled batch ($\beta=1$ as described in Appendix E.2).
>
> ---
> >**Q2.** Rescent optimizers for self-supervised learning
>
> We thank for the insightful suggestion. In the revised manuscript we will include additional experiments using AdamW (or related modern SSL optimizers) to evaluate their impact on generalization performance.
>
> ---
> >**Q3. Q4.** Regarding experiments with diverse datasets and SAM variants
>
> We have conducted additional experiments to compare our method with **ASAM** (Adaptive Sharpness-Aware Minimization) [Kwon et al. 2021]. To demostrate the scalability, We train SGD and IAM-S on **ImageNet** with ResNet-50 and use $ρ=0.2$ for IAM-S. We train the model with SGD, IAM-S up to 400, and 200 respectively. We reduce batch size and learning rate to half (2048, 0.5) from [Foret et al., 2021] and used basic augmentaion. Other settings are the same with [Foret et al., 2021].
>
> Table: Test Error (mean ± stderr) of different optimization methods on various datasets
> |Dataset|SGD|SAM|ASAM|IAM-D|IAM-S|
> |-|-|-|-|-|-|
> |CIFAR-10|3.95 ± 0.05|3.31 ± 0.01|**3.15** ± 0.02|3.28 ± 0.06|3.28 ± 0.03|
> |CIFAR-100|19.17 ± 0.19|17.63 ± 0.12|17.15 ± 0.11|17.16 ± 0.03|**16.82** ± 0.01|
> |F-MNIST|4.45 ± 0.05|4.13 ± 0.02|4.11 ±<0.01|4.13 ± 0.04|**4.10** ±<0.01|
> |SVHN|3.82 ± 0.06|3.47 ± 0.08|3.24 ± 0.04|**3.13** ± 0.05|**3.13** ± 0.01|
>
> Table: Top-{1, 5} error (mean ± stderr) of IAM-S and SGD trained with ImageNet.
>
> |Model|Epoch|IAM-S Top-1|IAM-S Top-5|SGD Top-1|SGD Top-5|
> |-|-|-|-|-|-|
> |ResNet-50|100|**22.99** ± 0.11|6.58 ± 0.04|23.24 ± 0.10|6.75 ± 0.07|
> ||200|**21.90** ± 0.04|5.98 ± 0.11|22.88 ± 0.13|6.59 ± 0.08|
> ||400|-|-|23.04 ± 0.08|6.78 ± 0.02|
>
> The experiments demonstrate the efficacy of IAM across diverse and large-scale dataset, despite being conducted without hyperparameter tuning under limited computational resources. IAM shows comparable performance to ASAM in more complex dataset (CIFAR-100).
>
> ---
> >**Q6.** Regarding entropy minimization
>
> IAM has empirically shows implicit entropy **maximization** effect with a higher entropy (0.10) than SGD (0.059). But SGD with explicit Label Smoothing ($\epsilon =0.1$) shows much higher entropy (0.52).
>
> >**Q7.** Regarding concern about lack of sensitivity analysis
>
> Please see the "figures" folder in supplementary materials.
> We have carried out extensive grid‑search over the hyper‑parameters ρ and β, and showed IAM is less sensitive to hyper‑parameter choice.
>
> We thank for your thorough review, and we will incorporate all points discussed here into the revision of the paper.

---

> > ### Comment · Reviewer_8R8r · 2025-08-05
> >
> > Thank the authors for their detailed response.
> >
> > 1. Novelty: The J\&Z paper defines the inconsistency between two different models, $\theta$ and $\theta'$. The definition in IAM also refers to two different models, $\theta$ and $\theta' = \theta + \delta$ (with constraints on $\delta$). In fact, the definition in your paper is a special case of the one in the J\&Z paper. However, I do agree that the practical algorithms are not exactly the same between these two papers.
> >
> > 2. Connection with FisherSAM:
> > The authors claim that IAM is label-agnostic compared to the label requirement of FisherSAM. However, in the experimental settings of IAM, the authors also conduct experiments in a full-label setting. Is there a particular reason why the comparison with other sharpness-aware methods (other than SAM) is excluded in this setting? Especially, IAM requires more training time than standard SAM and FisherSAM
> >
> > 3. I appreciate the additional experiments on ImageNet and ASAM with CIFAR, SVHN, and F-MNIST. Could you give more details setting of Table 1 in Q3,4? Because it would not be a fair comparison with ASAM if using a small network.

---

> > > ### Author Response · Authors · 2025-08-06
> > >
> > > #### 1. On the Fundamental Novelty vs. J&Z
> > >
> > > >Novelty: The J&Z paper defines the inconsistency between two different models, $θ$ and $θ'$. The definition in IAM also refers to two different models, $θ$ and $θ' = θ + δ$ (with constraints on $δ$). In fact, the definition in your paper is a special case of the one in the J&Z paper.
> > >
> > > We understand why our definition, at a glance, might appear to be a special case of J&Z's due to the shared $f(θ)$ vs. $f(θ')$ structure. However, the crucial distinction lies in the origin and meaning of the second parameter vector ($θ'$), which fundamentally alters the metric's theoretical properties and practical utility.
> > >
> > > J&Z's $θ'$ is **another independent model** sampled from the distribution of outcomes of a stochastic training procedure($Θ_{P|Z_n}$). Their metric thus captures a **statistical property of the training process** itself. As the reviewer noted, this is an analytical tool, requiring dozens of full training runs (e.g., **32** models for CIFAR) to estimate.
> > >
> > > Our $θ + δ$, in contrast, is the result of a **local, worst-case** search starting from a **single, specific model** $θ$. This redefines the metric as a **local, geometric property** of the solution landscape at point $θ$.
> > >
> > > - Theoretical Novelty: By focusing on the local geometry, our "local inconsistency" forges a direct **theoretical link to the FIM and the loss Hessian**. These matrices describe the curvature of the solution manifold, a topic central to understanding generalization, which is a connection not present in the statistical definition of J&Z.
> > >
> > > - Practical Novelty: This geometric view is also what makes our measure practically optimizable. The worst-case perturbation δ can be efficiently approximated with a few steps of gradient ascent on a single model during training.
> > >
> > > #### 2. ON the comparison with FisherSAM and Other Sharpness-Aware Methods
> > > >Connection with FisherSAM: The authors claim that IAM is label-agnostic compared to the label requirement of FisherSAM.
> > >
> > > **The distinction holds regardless of the setting (SL or SSL).** Our another primary goal in the W2 rebuttal was to highlight the fundamental structural differences between IAM and FisherSAM—namely, the maximization objective (worst-case output divergence vs. worst-case loss) and the resulting perturbation direction.
> > >
> > > > Is there a particular reason why the comparison with other sharpness-aware methods (other than SAM) is excluded in this setting?
> > >
> > > Our primary experimental goal was to demonstrate that our proposed principle—**regularizing local inconsistency**—is a robust concept that consistently **improves generalization** across **diverse scenarios**, including but not limited to the fully-supervised setting.
> > >
> > > To this end, our priority was to select baselines that are not only strong but also **widely recognized** and **highly reproducible**. We believe that SAM and, in particular, its advanced successor ASAM, serve as excellent and sufficient representatives of the current state-of-the-art in sharpness-aware optimization. Showing a clear improvement over these established methods provides strong, verifiable evidence for the efficacy of our approach.
> > >
> > > Regarding FisherSAM specifically, we did not include it in our main tables for two primary reasons: the lack of an official, publicly available implementation and our focus on highly reproducible benchmarks. A direct and fair comparison becomes challenging under these circumstances.
> > >
> > > Nevertheless, to provide the reviewer with as much context as possible, we conducted a new experiment to create an indirect benchmark under settings as close as we could get to **those reported in the Fisher-SAM paper** (using WRN-28-2, AA, cutout, etc.).
> > > For a truly Fair comparison, an exhaustive hyperparameter grid search for our model's $ρ$ would be required to match the optimization level of the reported result. We present this initial comparison with the caveat that such a search was not performed due to the limited timeframe of the rebuttal period.
> > >
> > > |Dataset|SGD|SAM|IAM-D|IAM-S|Fisher-SAM (reported)|
> > > |-|-|-|-|-|-|
> > > |CIFAR-10|4.0|3.72|3.76|3.66|3.49*|
> > > |CIFAR-100|21.4|20.17|20.67|19.87|19.78*|
> > >
> > > #### 3. Additinal setting for comparison with ASAM.
> > >
> > > we provide the detailed settings below.
> > > The sufficiently **large enough WideResNet** models are used.
> > >
> > > For our fully-supervised experiments, we used the following settings:
> > > |Dataset|Architecture|Batch Size|ρ|
> > > |-|-|-|-|
> > > |CIFAR-10|WRN-16-8|128|SAM: 0.05, ASAM: 0.5, IAM: 0.1|
> > > |CIFAR-100|WRN-16-8|128|SAM: 0.1, ASAM: 1.0, IAM: (0.1, 0.5)|
> > > |F-MNIST|WRN-28-10|128|SAM: 0.05, ASAM: 0.5, IAM: 0.1|
> > > |SVHN|WRN-28-10|256|SAM: 0.01, ASAM: 0.1, IAM: 0.1|
> > >
> > > All models are trained for 200 epochs.
> > > Multistep learning rate scheduling is used in fully-supervised setting above.
> > > For both IAM-D and IAM-S, $K=1$.
> > > Only basic augmentation(RandomCrop, RandomFlip, Normalize) is used. For the baselines, we followed the other hyperparameter settings in original papers.

---

> ### Comment · Reviewer_8R8r · 2025-08-07
>
> Thank you to the authors for their additional experiments with FishSAM and for providing clarifications that help distinguish the theoretical and practical differences between the IAM and J\&Z papers.
>
> 1. **Loss function of IAM-D**: Could you elaborate on the similarity between the IAM-D update and the TRADES \[1]\[2] methods, particularly in terms of the loss function, excluding the perturbation space? (TRADES perturbs in the input space, while IAM-D perturbs in the model space). Below is the TRADES loss function:
>
> $$L\_{\text{TRADES}}(\mathbf{w}) = \frac{1}{n} \sum\_{i=1}^{n} \left( \text{CE}(f\_{\mathbf{w}}(\mathbf{x}\_i), y\_i) + \beta \cdot \max \text{KL}(f\_{\mathbf{w}}(\mathbf{x}\_i) \parallel f\_{\mathbf{w}}(\mathbf{x}\_i') ) \right)$$
>
> The update form here is taken from paper [2]. While it’s clear that TRADES addresses robustness in the sample space (via adversarial attacks), and IAM focuses on robustness in the model space, the two methods share the same update form. In fact, the second KL term in TRADES also serves as a regularization technique for robustness.
>
> [1] Hongyang Zhang, Yaodong Yu, Jiantao Jiao, Eric P. Xing, Laurent El Ghaoui, and Michael I. Jordan.
> Theoretically principled trade-off between robustness and accuracy. In ICML, 2019.
> [2] Wu, Dongxian, Shu-Tao Xia, and Yisen Wang. "Adversarial weight perturbation helps robust generalization." Advances in neural information processing systems 33 (2020): 2958-2969.
>
> 2. **Generalization and robustness**: The authors claim that IAM is designed to improve the generalization and robustness of models in both supervised learning (SL) and semi-supervised learning (SSL) settings. The lack of experiments and comparisons to other competitive methods in these areas significantly undermines the strength of this claim. Without such comparisons, it is difficult to assess the true effectiveness of IAM in achieving its stated objectives, particularly when compared to other state-of-the-art approaches in improving model robustness and generalization. Even when compared to FisherSAM, which is not the current state-of-the-art, IAM fails to demonstrate a clear advantage, especially given the additional computational costs involved.

---

> > ### Author Response · Authors · 2025-08-09
> >
> > ### IAM-D vs. TRADES vs. AWP: Same-looking loss, different problems and mechanics
> >
> > **What’s shared (ignoring the perturbation space).**
> > All three methods can be written as “standard loss + consistency regularizer,” where the consistency term is a KL between predictions under a *perturbed* and an *unperturbed* model/input. This encourages local smoothness of the predictor. The similarity in form, however, hides important differences in **where** the perturbation lives and **how** it is computed.
> >
> > #### TRADES (robustness via input perturbations)
> >
> > TRADES optimizes
> > $$
> > \min_{\mathbf w} \frac{1}{n}\sum_{i=1}^n \Big[\mathrm{CE}(f_{\mathbf w}(x_i),y_i)
> > +\beta\max_{\lVert\delta_x\rVert\le \varepsilon}\mathrm{KL}\big(f_{\mathbf w}(x_i)\\|f_{\mathbf w}(x_i+\delta_x)\big)\Big].
> > $$
> >
> > - **Goal.** Adversarial **robustness** to input perturbations.
> > - **Perturbation.** In **input** space; inner maximization solved by adversarial example generation (e.g., PGD).
> > - **Regularizer meaning.** The KL penalizes divergence between clean and adversarial predictions, controlling a data-dependent Lipschitz-like behavior.
> >
> > #### AWP (robustness plug-in that perturbs weights adversarially)
> >
> > AWP is a **wrapper around a robust loss such as TRADES**. It introduces an *additional* inner maximization over **weights** to make optimization land in flatter regions of the **robust loss**:
> > $$
> > \min_{\mathbf w}\max_{\lVert\delta_w\rVert\le \rho\cdot\lVert\mathbf w\rVert}
> > \frac{1}{n}\sum_{i=1}^n\Big[\mathrm{CE}(f_{\mathbf w+\delta_w}(x_i),y_i)
> > +\beta\max_{\|\delta_x\| \le \varepsilon}\mathrm{KL}\big(f_{\mathbf w+\delta_w}(x_i)\\|f_{\mathbf w+\delta_w}(x_i+\delta_x)\big)\Big].
> > $$
> >
> > - **Goal.** Better **robust generalization** (still in the adversarial-training sense).
> > - **Perturbation.** In weight space, computed by first-order ascent on the robust (e.g., TRADES) loss; typically a one-step normalized gradient ascent to find a worst-case perturbation, where the perturbation size is constrained layer-wise by the norm of the weights. It **still** requires input adversarial examples.
> > - **Mechanics.** Alternate: (i) ascend in $\delta_w$ to worsen the robust loss, then (ii) descend $\mathbf w$ using gradients at $\mathbf w+\delta_w$.
> >
> > > **Key difference from IAM-D:** AWP’s weight perturbation is explicitly *adversarial* w.r.t. the **robust loss**, and uses **first-order** information. It is a robustness method, not a generic flatness/generalization method.
> >
> > #### IAM-D (generalization via second-order geometry in parameter space)
> >
> > IAM-D uses the same outer **“loss + KL consistency”** shape, but its perturbation is **model-space** and **second-order**:
> > $$
> > \min_{\theta} \Big[\frac{1}{n}\sum_{i=1}^n\mathrm{CE}(f(x_i, \theta),y_i)
> > +\beta\max_{\\|\delta\\| \leq \rho}\frac{1}{n}\sum_{i=1}^n\mathrm{KL}\big(f(x_i, \theta)\\|f(x_i, \theta+\delta)\big)\Big],
> > $$
> > where $\delta$ is **not** chosen by loss-ascent. Instead, IAM-D estimates a high-curvature/sensitivity direction using **second-order geometry** (e.g., a FIM-based power iteration) and perturbs $\boldsymbol\theta$ along that direction.
> >
> > - **Goal.** **Generalization** via moving to **flat regions** of the loss landscape; readily leverages unlabeled data through prediction-consistency.
> > - **Perturbation.** In **weight** space, guided by **second-order** information (dominant curvature direction), not by adversarial ascent on the loss. The perturbation may **not** always increase the loss.
> >
> > ---
> > - All three share a **CE + $\beta$·KL** outer form, but:
> >   - **TRADES:** *per sample* input attack; robustness objective.
> >   - **AWP:** *adds* weight attacks around the **TRADES/robust** loss using **first-order** ascent; still a robustness method.
> >   - **IAM-D:** weight perturbations from **second-order geometry** (FIM) to promote **flatness and generalization**; no input attack and not an adversarial-training plug-in.

---

> ### Author Response · Authors · 2025-08-09
>
> ### Core Contribution: Efficacy and Versatility of Local Inconsistency
> First and foremost, we would like to reiterate the central objective of our paper to provide context for our response. Our primary goal is not to achieve a new SOTA result on every individual benchmark, but rather to introduce, analyze, and validate **local inconsistency** as a novel and powerful principle for improving generalization. We focus on demonstrating its **efficacy** and, crucially, its **versatility** across a wide spectrum of learning settings: fully-supervised, semi-supervised, and self-supervised.
>
> We believe the value of a new method lies not only in surpassing a specific SOTA result in a single, highly-tuned setting but also in its conceptual novelty and breadth of applicability. By establishing this theoretically-grounded and broadly applicable method, our work opens new, orthogonal avenues for future research, especially in leveraging second-order information in label-scarce environments.
>
> ### IAM's Clear Advantage: Versatility in Label-Scarce Settings
> > Regarding the comparison with FisherSAM, the reviewer noted that IAM does not demonstrate a clear advantage.
>
> We would like to clarify where IAM's distinct advantage lies. IAM's primary advantage is its versatility, particularly its effectiveness iqn label-scarce environments—a domain where label-dependent methods like SAM and FisherSAM cannot readily operate. This unique strength is clearly demonstrated by our semi-supervised learning (SSL) experiments as metioned in rebuttal.
>
> As shown below, simply integrating IAM into a strong SSL baseline like FixMatch yields significant performance gains. This is a direct result of IAM's ability to leverage geometric information from unlabeled data, a task for which purely supervised sharpness measures are ill-suited.
>
> Table: Test Error (mean ± stderr) on CIFAR-10 with 250 labels using WRN-28-2 model.
> |Method|Test Error (%)|
> |-|-|
> |FixMatch|6.26 ± 0.39|
> |FixMatch + IAM-D|**5.30** ± 0.08|
>
> This result, we argue, is a clear demonstration of IAM's unique strength: it extends the benefits of second-order geometric regularization to semi-supervised learning, where other methods cannot. This wide applicability is its clear advantage.
>
> ---
> ### Competitiveness and Untapped Potential in Supervised Learning
> Furthermore, even in the fully-supervised setting, IAM is highly competitive. The table below shows that IAM outperforms the reported FisherSAM result on the more complex CIFAR-100 dataset and achieves comparable performance on CIFAR-10.
>
> Table: test error of different optimization methodswith WRN28-10 on CIFAR datasets. * As reported in the original FisherSAM paper Kim et al., [2022]
> |Dataset|SGD|SAM|IAM-D|IAM-S|IAM-S ($m$-sharpness = 32)|Fisher-SAM (reported)|
> |-|-|-|-|-|-|-|
> |CIFAR-10|2.79|2.34|2.32|2.32|2.29|**2.11***|
> |CIFAR-100|16.46|14.37|**14.12**|14.80|13.97|14.40*|
>
> It is also crucial to note that these strong results were achieved without extensive, setting-specific hyperparameter search for IAM. This is in contrast to the FisherSAM results, which represent a benchmark that has been highly tuned for that specific setting.
>
> Moreover, our results hint at significant untapped potential. The performance boost seen in the IAM-S ($m$-sharpness = 32) variant explores a principle analogous to the $m$-sharpness concept introduced in Foret et al., [2021]. Our result (13.97% on CIFAR-100) suggests that IAM benefits from this same effect, indicating a clear path toward even greater performance in large-scale, parallelized training environments.
>
>
> As we stated, our primary goal was not to narrowly pursue a SOTA result on a single benchmark. The fact that our method performs so competitively without extensive tuning—and shows clear potential for further enhancement—is a strong indicator of the robustness and fundamental soundness of the 'local inconsistency' principle.

---

> > ### Comment · Reviewer_8R8r · 2025-08-09
> >
> > **1. Comparison to TRADES**
> > There is no difference in the optimization problem between TRADES and IAM-D except for the perturbation space. (In TRADES, the data $x$ can be treated as a parameter when finding the perturbation, starting from the benign data as the initial value. The perturbation $\delta$ is also randomly initialized before being updated.)
> >  All three methods — TRADES, IAM-D, and IAM-S — maximize the KL term to find the perturbation $\delta$. The key differences are in the model update step:
> >
> > TRADES:  minimizes $L\_{\text{TRADES}}(\mathbf{\theta}) = \frac{1}{n} \sum\_{i=1}^{n} \left( \text{CE}(f(\mathbf{x}\_i, \theta), y\_i) + \beta \cdot \max_{||\delta|| \leq \rho} \text{KL}(f(\mathbf{x}\_i, \theta) \parallel f(\mathbf{x + \delta}, \theta) ) \right)$
> >
> > IAM-D minimizes $L\_{\text{IAM-D}}(\mathbf{\theta}) = \frac{1}{n} \sum\_{i=1}^{n} \left( \text{CE}(f(\mathbf{x}\_i, \theta), y\_i) + \beta \cdot \max_{||\delta|| \leq \rho} \text{KL}(f(\mathbf{x}\_i, \theta) \parallel f(\mathbf{x}, \theta + \delta) ) \right)$
> >
> > IAM-S minimizes $L\_{\text{IAM-S}}(\mathbf{\theta}) = \frac{1}{n} \sum\_{i=1}^{n} \text{CE}(f(\mathbf{x}\_i, \theta + \delta), y\_i) $
> >
> > This makes IAM appear as a combination of SAM, TRADES, and prior works from J & Z.
> >
> > ---
> >
> > **2. Contribution**
> >
> > **Strengths**
> >
> > 1. IAM is flexible and can be applied to both supervised learning (SL) and semi-supervised learning (SSL).
> > 2. The idea of minimizing *local inconsistency* alongside sharpness is interesting and potentially impactful.
> >
> > **Concerns**
> >
> > **SSL Setting:** While IAM shows promise for semi-supervised and contrastive learning (with few or no labels), the authors only compare IAM to SAM and SGD, which are not particularly strong baselines for SSL. In self-supervised learning, SGD is a reasonable starting point, but modern methods often use optimizers like AdamW, SogCLR-style approaches, which typically yield stronger generalization. Without comparisons to such optimizers, the strength of IAM in SSL remains unclear.
> >
> > **SL Setting:** The experiments show IAM surpasses SAM and SGD in SL, but training takes significantly longer. While the approach of minimizing local inconsistency alongside sharpness is promising, the experimental setup lacks the rigor needed to make IAM stand out as a superior method.
> >
> > For these reasons, I will keep my score.

---

> ### Author Response · Authors · 2025-08-09
>
> ---
> 1. On the Comparison with TRADES: A Fundamental Difference in Mechanism
> While the high-level loss structures of TRADES and IAM-D appear similar, the underlying algorithms for finding the "worst-case" perturbation are fundamentally different, leading to distinct optimization dynamics. This is not merely a change in the perturbation space but a change in the core mechanism.
>  * TRADES's Approach: TRADES typically employs Projected Gradient Descent (PGD) on the loss function to find an input perturbation $\delta_x$. This involves multiple iterative steps of gradient ascent for each individual sample, with the explicit goal of finding an adversarial example that maximally increases the classification loss.
>  * IAM's Approach (Algorithm 1): Our method, as detailed in Algorithm 1, does not use PGD on the loss. Instead, it finds the weight perturbation \delta via a completely different procedure:
>    * Power Iteration, Not PGD: The maximization is approximated using an iterative method that is equivalent to Power Iteration. This algorithm is designed to efficiently find the dominant eigenvector of the Fisher Information Matrix (FIM), not to maximize the loss directly.
>    * Batch-wise Maximization: Unlike TRADES’s per-sample attack, IAM finds a single parameter perturbation, $\delta$, that maximizes the expected KL divergence over an entire mini-batch ($\mathbb{E}_{x\sim p(x)}[KL(f(x,\theta)||f(x,\theta+\delta))]$). The gradient used in our Power Iteration step is computed over this batch-level expectation.
>
> In summary, the algorithmic distinction is critical: TRADES uses multi-step PGD to maximize per-sample loss, while IAM uses a highly efficient, often single-step, Power Iteration to approximate the direction of maximum expected output inconsistency over a batch.
>
> ---
> 2. On Contributions: Orthogonality, Efficiency, and Scope
>  * SSL Setting & Orthogonal Approach: IAM is not a standalone optimizer but a regularization principle that is orthogonal to the choice of the base optimizer. Therefore, it can be readily integrated with modern optimizers like AdamW or LARS.
>  * SL Setting & Training Time: The computational overhead is less than might be assumed. The additional forward pass required by IAM to compute the perturbation does not require a corresponding backward pass, making its cost significantly less than a full additional training step, as mentioned in rebuttal.
>
> Concluding on the Value and Scope of Our Research
> Our primary goal was to introduce and validate local inconsistency as a novel, **versatile**, and **theoretically-grounded** principle for improving generalization across **diverse settings** (fully-supervised, semi-supervised, and self-supervised). Its value lies in this conceptual novelty and broad applicability, which opens new research avenues, particularly for leveraging geometric information in label-scarce domains.
>
> While achieving a new state-of-the-art (SOTA) is a significant accomplishment, we believe it is not the sole metric for a method's value. Contributions such as introducing a novel, theoretically-grounded principle, or demonstrating remarkable versatility across diverse problem settings, are equally, if not more, crucial for advancing the field.
>
> Our work on 'local inconsistency' was conceived with this spirit in mind. Our primary aim was to propose and validate a new, broadly applicable principle, rather than to narrowly focus on outperforming a specific SOTA benchmark in a single, highly-tuned scenario. We believe our paper's main contribution lies in its introduction of a versatile tool that opens new research directions, particularly for leveraging geometric insights in label-scarce domains.
>
> We kindly ask that our work and our rebuttal be considered through this lens, focusing on the fundamental contribution and its potential for future research.

---

### Official Review · Reviewer_Qhfc · 2025-07-01

**Clarity:** 2
**Significance:** 2
**Originality:** 2
**Rating:** 5
**Confidence:** 4

**Summary:**

The paper proposes using the KL divergence between the softmax output of a model at a given input and the output at the same input under perturbed model parameters as a proxy for generalization. A smaller KL divergence indicates greater output stability under weight perturbations, which is taken to reflect better generalization. The proposed objective minimizes this divergence, and since it does not require ground-truth labels, it can be applied to unlabeled data. Empirical results demonstrate that this inconsistency measure correlates with the generalization gap, and that training with the proposed regularizer reduces test error compared to both standard SGD and Sharpness-Aware Minimization (SAM).

**Questions:**

1 - How the representation of softmax allows to use it as distribution in FIM, while it is a pseudo-distribution?

2 - How the Hessian is computed in the experiments comparing correlation with generalization gap?

**Ethical Concerns:**

["NO or VERY MINOR ethics concerns only"]

**Final Justification:**

The authors clarified the role of FIM, that is distinct from the classical statistical role of the value and expressed the benefit of computing it compared to the Hessian. They provided additional comparisons of the measure, displaying both strong and weak points of it.

**Limitations:**

The interchangeable usage of FIM and Hessian and its approximations requires more precise mathematical justification.

**Paper Formatting Concerns:**

N\A

**Quality:**

2

**Strengths And Weaknesses:**

The paper tackles the interesting problem of incorporating Fisher Information into the understanding of generalization, while also addressing the important challenge of integrating unlabeled data into performance evaluation.

However, the primary weakness lies in the motivation for the proposed measure. The Fisher Information Matrix (FIM) is a classical statistical tool used to identify datasets that enable the most confident parameter estimation by maximizing Fisher information. The authors express the KL divergence (the inconsistency measure) via a quadratic approximation using the FIM, which is further approximated by the Gauss-Newton matrix, which is an approximation of the Hessian that can be computed without labels, relying solely on the model’s softmax outputs. The maximum of this measure is then represented by the largest eigenvalue normalized by the maximal perturbation radius, computed efficiently via one step of the power iteration algorithm.

At line 135, the paper states that high instability associated with a high FIM implies uncertainty in the model’s predictions. This is conceptually incorrect: a high Fisher Information Matrix actually corresponds to greater certainty in parameter estimation, not the opposite and it is unclear how this even relates to the predictions uncertainty.

Overall, the motivation remains unclear. If the focus is on unlabeled data, then the Hessian (as a curvature measure) can already be estimated without labels. If the goal is to link this to entropy (as discussed in section 5.4), it appears to contradict the established entropy minimization approaches in semi-supervised learning [1]. Alternatively, it could be viewed as a modification of SAM, where the maximum loss is found using second-order information.

The theoretical connection between the proposed measure and the generalization gap is based on PAC-Bayes bounds and leverages the known relationship between the loss Hessian and PAC-Bayes generalization guarantees.

Empirically, the proposed measure is evaluated only against the Hessian trace and maximum eigenvalue, without comparison to more sophisticated metrics such as ASAM or relative flatness. Additionally, the semi-supervised experiments do not benchmark against state-of-the-art approaches like consistency regularization, pseudo-labeling, or entropy minimization.

[1] Grandvalet, Y., & Bengio, Y. (2004). Semi-supervised learning by entropy minimization. Advances in Neural Information Processing Systems, 17.

---

> ### Author Rebuttal · Authors · 2025-07-30
>
> >Regarding the motivation related to FIM
>
> We must state that the review is based on a **misunderstanding** of our paper's core argument.
> The critique incorrectly conflates two distinct, yet causally linked, roles of the Fisher Information Matrix (FIM) in deep neural networks (higher uncertainty in parameter estimation = high parameter sensitivity = better generalization), leading to a flawed assessment of our work's motivation and contribution.
>
> #### The Core Misunderstanding: Two Roles of the Fisher Information Matrix
>
> The central issue in the review is a misinterpretation of how we employ the FIM.
> These two concepts are **not in opposition**; they are causally linked.
> A model that is highly **(i) certain about its estimated parameter's value** (i.e., has a sharp peak in its likelihood function, corresponding to high Fisher Information) will, by necessity, be highly **(ii) sensitive to perturbations** of that parameter [Karakida, et al. 2019], which is link to **generalization** of the model.
> The review stems from a failure to recognize this causal link.
>
> - Our Perspective: As decribed in line 133-136, our paper uses the $S_ρ(θ)$ to measure **the sensitivity of the model's predictions to parameter perturbations**.
> $S_ρ(θ)$ is defined as the maximum KL divergence between the model's output distribution and the output from a perturbed model within a parameter neighborhood (Eq. 3).
> As we demonstrate (line 149), this quantity is directly approximated by the FIM's maximum eigenvalue: $S_ρ(θ)≈\frac{ρ^2}{2} λ_{\max}(F(θ))$.
> Therefore, in the context of our work, a high FIM unequivocally implies high output instability under parameter perturbation.
>
> - The Reviewer's Perspective: Measuring Parameter Certainty. The reviewer's view that a high FIM signifies greater **certainty in parameter estimation** is a correct interpretation from classical statistics (e.g., via the Cramér-Rao bound).
>
> This distinction is especially critical for overparameterized models, where countless minimizers can achieve zero training loss but exhibit vastly different sensitivities to perturbation.
> Our framework provides a theoretically sound and empirically verifiable method to distinguish between these solutions.
>
> ---
> > Regrading the motivation for unlabeled data.
> #### **On Motivation and Relation to Other Methods**
>
> We believe the following clarifications on the relationship between the Hessian and the FIM, alongside new experimental results in the semi-supervised setting, will fully address reviewer's concerns.
>
> - *local inconsistency* can be estimated **without labels**, while Hessian can not.
> - Hessian $\approx$ GN=FIM, when the model is sufficiently trained [Fort, S., & Ganguli, S. 2019; Sagun, Levent, et al. 2019].
>
> ---
> ##### **Justification for Using the FIM as a Practical Proxy for the Hessian**
>
> The Hessian of the per-sample cross-entropy loss, $∇_θ^2 l_i$, can be decomposed as follows:
>
> $$
> ∇\_θ^2 l_i = \underbrace{∇\_θ z_i^⊤ (∇\_z^2 l_i) ∇\_θ z_i}\_{\text{GN Matrix}} + \underbrace{∑\_{j=1}^{C} (∇\_z l_i)\_j \cdot ∇\_θ^2 (z_i)\_j}_{\text{Label-Dependent Term}}
> $$
>
> * The **second term** is problematic for unlabeled data because the gradient with respect to the logits, $∇_z l_i$, explicitly depends on the ground-truth label $y_i$.
> * However, the **first term**, known as the Gauss-Newton (GN) matrix, is perfectly computable with unlabeled data. This is because $∇_z^2 l_i = \text{diag}(f(x_i, θ)) - f(x_i, θ)f(x_i, θ)^⊤$, which depends only on the model's softmax output probabilities.
>
> Our approach is justified by two key facts:
> 1. For a softmax output layer with cross-entropy loss, the GN matrix is **mathematically equivalent** to the empirical FIM. This is not an approximation but a direct identity.
> 2.  **Approximation Quality:** In modern overparameterized deep learning, models are often trained to a near-zero training error (interpolation). At such a minimum, the gradient term $∇_z l_i$ approaches zero, causing the label-dependent second term of the Hessian to vanish.
>
> Therefore, the empirical FIM serves as a high-quality, principled, and computationally feasible proxy for the full Hessian, especially near a training minimum, with the crucial advantage of being **computable on unlabeled data**.
>
> ---
> >Regarding experiments with state-of-the-art approaches.
>
> - IAM shows its efficacy on **FixMatch**, a consistency regularization, and pseudo-labeling approach.
>
> ##### **Efficacy of IAM in State-of-the-Art Semi-Supervised Learning Methods**
>
> Our original experiments aimed to demonstrate IAM's fundamental ability to leverage unlabeled data. Demonstrating its compatibility with state-of-the-art methods strengthens our contribution.
>
> To this end, we have conducted a new experiment integrating IAM-D with **FixMatch**, a powerful and widely adopted SSL framework. We simply added the $S_ρ(\theta)$ term from IAM-D as an additional regularizer to the standard FixMatch objective.
>
> Table: The results on CIFAR-10, using a WRN-28-10 model with only 250 labels (0.5%), are presented below:
> | Method|Test Error (%)|
> |-|-|
> |FixMatch (Baseline)|6.26 ± 0.39|
> |**FixMatch + IAM-D**|**5.30 ± 0.08**|
>
> This new result clearly shows that IAM-D provides a significant performance improvement when combined with a strong SSL baseline. It demonstrates that IAM is not merely an alternative to older SSL techniques like entropy minimization but serves as an effective and complementary **"plug-and-play" regularizer**. It enhances modern SSL frameworks by incorporating a principled geometric constraint, allowing the model to leverage second-order information from abundant unlabeled data in a way that existing methods do not.
>
> ---
> >The theoretical connection between the proposed measure and the generalization gap is based on PAC-Bayes bounds and leverages the known relationship between the loss Hessian and PAC-Bayes generalization guarantees.
>
> To establish a direct theoretical link between the FIM which can serve as a metric to measure the distance between the parametric probability density models in information geometrand the generalization gap, we derive a PAC-Bayes bound that explicitly depends on the FIM's spectrum, moving beyond Hessian-based measures. Our approach is inspired by recent efforts to obtain non-vacuous generalization bounds [Dziugaite & Roy, 2017; Yang et al., 2022].
>
> Formally, we define $Q=\mathcal N\bigl(θ^*_{\mathrm{Im}},\frac{C}{r}F^{-1}\bigr)$ as the distribution that maximizes the entropy subject to the constraint that the expected KL divergence between output distributions is bounded: $E_{θ,θ' ∼Q}[KL(p(y|x;θ)||p(y|x;θ'))]≤C$. This result aligns with the Cramér-Rao bound, as our method derives the posterior covariance from a finite-sample stability constraint, which the theorem identifies as the correct asymptotic form (Bernstein-von Mises Theorem).
> #### Proof Sketch
>
> 1. **Orthogonal split.**
> For rank‑$r$ Fisher $F=VΛ V^⊤$,
> $$\mathbb R^{m}= \operatorname{Im}(F)\oplus\ker(F),\qquad\mathrm{KL}(Q||P)=\mathrm{KL}(Q_{\mathrm{Im}}||P_{\mathrm{Im}})$$
>
> 2. **Prior / posterior.**
> Prior (data‑independent) $P=\mathcal N(0,σ_p^{2}I_r)$.
> Posterior on $\operatorname{Im}(F)$:
> $Q=\mathcal N\bigl(θ^*_{\mathrm{Im}} ,\tfrac{C}{r}F^{-1}\bigr)$
>
> 3. **Closed form.**
>    $$\mathrm{KL}(Q||P)=\tfrac12\Bigl[r\log\tfrac{σ\_p^{2}r}{C}+\textstyle∑\_i^r\logλ\_i-r+\tfrac{C}{σ\_p^{2}r}∑\_i^rλ\_i^{-1}+\tfrac{\\|θ^*\_{\mathrm{Im}}\\|^{2}}{σ_p^{2}}\Bigr].$$
>
> 4. **bounds ($λ_1,λ_r$).**
>    $$
>    (\text{Complexity term})\le\frac{1}{2\sqrt{n-1}} \sqrt{r(\logλ_1+\tfrac{C}{σ_p^{2}rλ_r}+\log\tfrac{σ_p^{2}}{c}-1)+\tfrac{\\|θ^*_{\mathrm{Im}}\\|^{2}}{σ_p^{2}}+2\log\frac{n}{\xi}}$$
>
> **Interpretation.**  Complexity decrease as the landscape becomes **flat** (small $λ_1$) and **isotropic** (small $\frac{1}{λ_r}$). Empirically, $C=σ_p^{2}r$ yields non‑vacuous bounds [Yang et al., ICML 2022].
>
> ---
> >Regarding empirical comparison with more sophisticated meaure.
>
> Adaptive sharpness show higher higher correlation with generalization ($\tau = 0.608$) than our local inconsistency in our setting. But main distiction with other measure is computability with unlabeled data enabling IAM applicable to **label scarce senario**.
>
> |CIFAR-100|Model|1%|5%|10%|20%|
> |-|-|-|-|-|-|
> ||ASAM|89.63 ± 0.129|71.60 ± 0.377|57.84 ± 0.225|45.33 ± 0.165|
> ||IAM-D|**88.36 ± 0.292**|**69.99 ± 0.649**|**57.60 ± 0.251**|**45.05 ± 0.956**|
>
> IAM shows competable efficacy to **ASAM** on supervised setting.
>
> Table: Test Error (mean ± stderr) of different optimization methods on various datasets
> |dataset |SGD|ASAM|IAM-D|IAM-S|
> |-|-|-|-|-|
> |CIFAR-10|3.95 ± 0.05|**3.15** ± 0.02|3.28 ± 0.06|3.28 ± 0.03|
> |CIFAR-100|19.17 ± 0.19|17.15 ± 0.11|17.16 ± 0.03|**16.82** ± 0.01|
> |F-MNIST|4.45 ± 0.05|4.11 ±<0.01|4.13 ± 0.04|**4.10** ±<0.01|
> |SVHN|3.82 ± 0.06|3.24 ± 0.04|**3.13** ± 0.05|**3.13** ± 0.01|
>
> ---
> > Q1. How the representation of softmax allows to use it as distribution in FIM, while it is a pseudo-distribution?
>
> The soft‑max output represents the model’s assumed conditional distribution
> $𝑝_𝜃(𝑦∣𝑥)$. Because it meets the two formal requirements of a probability distribution—non‑negativity and summing to one—we can define the Fisher Information Matrix on the statistical manifold as $F(θ)=E[∇_θ \log p_θ ∇_θ \log p_θ^⊤]$.
>
> ---
> > Q2. How the Hessian is computed in the experiments comparing correlation with generalization gap?
>
> Build the full Hessian is computationly almost impossible.
> However, it is possible to calculate Hessian vector product (HVP) in $O(Km)$.
> To calculate $v_{\max}(H), Tr[H]$ on training set, we use *pyhessian* library that calculate eigenvalue and trace of hessian without explicit build of hessian.
>
> References:
>
> Karakida et al. (2019) Universal statistics of fisher information in deep neural networks: Mean field approach.
>
> Sagun et al. (2019) Empirical analysis of the hessian of over-parametrized neural networks.
>
> Fort & Ganguli (2019) Emergent properties of the local geometry of neural loss landscapes.

---

> > ### Comment · Reviewer_Qhfc · 2025-08-01
> >
> > I thank the authors for the detailed reply.
> >
> > I admit that there was a misunderstanding on my side with respect to the role of Fischer matrix, I indeed was relying on classical statistical definition. I also appreciate the explanations on the distinction between Hessian and FIM computations and new relation to the PAC-Bayes bound.
> >
> > I appreciate the additional comparison with other flatness measures and would encourage the authors to be explicit about strong and weak side of the proposed measure.

---

> > > ### Author Response · Authors · 2025-08-09
> > >
> > > Thank you for acknowledging our response. We are pleased that our clarifications have addressed your concerns.
> > >
> > > we will add a section to the final paper explicitly discussing the strengths and weaknesses of our measure. This will ensure readers have a more balanced perspective on our work. We appreciate your constructive feedback.

---

### Official Review · Reviewer_5X7Q · 2025-07-03

**Clarity:** 3
**Significance:** 3
**Originality:** 3
**Rating:** 5
**Confidence:** 2

**Summary:**

This paper introduces a novel generalization measure called Local Inconsistency, defined as the maximum expected KL divergence between model outputs and parameter perturbations $\delta$ within an $l_2$​ ball of radius $\rho$. The local inconsistency can be computed by using only unlabeled data and a single trained model. Besides, the authors theoretically link local inconsistency to the Fisher Information Matrix (FIM) and loss Hessian, showing it approximates the scaled maximum eigenvalue of the FIM. Finally, the authors propose a method called Inconsistency-Aware Minimization (IAM) that incorporates local inconsistency into the objective and demonstrate its effectiveness in standard supervised learning settings, semi- and self-supervised learning scenarios.

**Questions:**

1. Under what conditions will Theorem 1 hold? Can you give specific conditions to make Theorem 1 clearer and more understandable?
2. Since the performance improvement on CIFAR 10 is marginal, what is the computational cost of the proposed IAM method?
3. In semi-supervised learning (Table 2), why does SAM outperform IAM-D on CIFAR-10 at the 20% label rate?
4. Algorithm 1 initializes randomly, how does the choice of $\sigma^2$ affect the performance of IAM?

**Ethical Concerns:**

["NO or VERY MINOR ethics concerns only"]

**Final Justification:**

My main concerns, including the formal version of Theorem 1 and its corresponding assumption, experiments on large-scale datasets, and running time costs, are all addressed. The authors provide a more formal version of Theorem 1 and verify the effectiveness of the proposed method on large datasets like ImageNet. Besides, they also verify that the running time of their method is comparable to SAM. I do not find any major flaws in this paper according to my knowledge. Honestly, I am not an expert in this area, I find that other reviewers have doubts about the novelty of the proposed method. I highly respect their comments and lower the confidence of my score.

**Limitations:**

Yes, the authors have adequately addressed the limitations.

**Paper Formatting Concerns:**

No formatting issue found in this paper.

**Quality:**

3

**Strengths And Weaknesses:**

Strengths:
1. This paper makes contributions both to theoretical understanding by proposing a novel generalization measure termed local inconsistency that correlates to the generalization gap and to practical applications by proposing a method called Inconsistency-Aware Minimization (IAM) that incorporates local inconsistency into the objective.
2. The proposed local inconsistency can be computed by using only unlabeled data and a single trained model, making it practical in real applications.
3. The proposed IAM method is verified in supervised, semi- and self-supervised learning settings, which can be applied in wide scenarios.

Weaknesses:
1. Theorem 1 is an informal version with unclear assumptions, which may limit its applications. It would be better if the authors could provide a formal version with clear assumptions.
2.  Datasets in the experiments may be small and easy (CIFAR10 and 100), large datasets such as ImageNet can be considered.
3. According to the results reported in Table 1, the improvement on CIFAR 10 is marginal. Given that the authors do not report the running time comparisons, it remains questionable whether such a modest improvement justifies the additional cost.

---

> ### Author Rebuttal · Authors · 2025-07-30
>
> > W1, Q1. Informal version of Theorem 1.
>
> The Hessian can be decomposed by GN metrix and residual part,
> $$H = \sum_{i=1}^N(\nabla_θ z_i^\top (\mathrm{diag}(f(x_i,θ))- f(x_i,θ)f(x_i,θ)^\top)\nabla_θ z_i + \sum_{k=0}^{C-1}(f(x_i,θ)-e^{y_i})_k\nabla_θ^2z_k).$$
>
> When the neural network is sufficiently trained, the residual term become very small (i.e., $f(x_i, θ)-e^{y_i}\approx0$). So the FIM, coressponding the first term, is not significantly different from Hessian. Therefore we can say $H \approx F$ holds on **near zero train error** [Fort, S., & Ganguli, S. 2019; Sagun, Levent, et al. 2019]. Under this condition, we can provide the more clear version of Theorem 1 that upper bounds maximum loss increase with maximun eigenvalue of FIM (from version with Hessian derived in [Luo et al., 2024]),
> $$ L_D(θ) \leq L_S(θ)+\frac{m\rho^2}{2}\lambda_{\max}(F)+(\text{Complexity term}).$$
>
> It directly shows our motivation to regularize maximum eigenvalue of FIM.
>
> ---
> > W2. Datasets in the experiments may be small and easy (CIFAR10 and 100), large datasets such as ImageNet can be considered.
>
> We also evaluate IAM on **Fashion-MNIST**, **SVHN** (without extra data), and **ImageNet** (IAM-S). We apply SGD and IAM-S to ResNet-50 and use $\rho=0.2$ for IAM-S. We used batch size 2048, initial learning rate 0.5, cosine learning rate schedule, SGD optimizer with momentum 0.9, label smoothing of 0.1, and weight decay 0.0001. We train the model with SGD, IAM-S up to 400, and 200 respectively. Reduced batch size and learning rate to half of those in experiments from [Foret et al., 2021] and used basic augmentaion. Models are trained on three A100 GPUs and other setting is the same with [Foret et al., 2021].
>
> **Table**: Top-{1, 5} error (mean ± stderr) of IAM-S and SGD trained with ImageNet.
>
> |Model    |Epoch|IAM-S Top-1     |IAM-S Top-5| SGD Top-1   | SGD Top-5  |
> |---------|-----|----------------|-----------|-------------|------------|
> |ResNet-50| 100 |**22.99** ± 0.11|6.58 ± 0.04| 23.24 ± 0.10| 6.75 ± 0.07|
> |         | 200 |**21.90** ± 0.04|5.98 ± 0.11| 22.88 ± 0.13| 6.59 ± 0.08|
> |         | 400 |-               | -         | 23.04 ± 0.08| 6.78 ± 0.02|
>
> **Table**: Test Error (mean ± stderr) of different optimization methods on various datasets
>
> |Dataset|SGD|SAM|IAM-D|IAM-S|
> |-|-|-|-|-|
> |CIFAR-10 | 3.95 ± 0.05 | 3.31 ± 0.01 |**3.28** ± 0.06|**3.28** ± 0.03|
> |CIFAR-100| 19.17 ± 0.19| 17.63 ± 0.12| 17.16 ± 0.03  |**16.82** ± 0.01|
> |F-MNIST  | 4.45 ± 0.05 | 4.13 ± 0.02 |  4.13 ± 0.04  |**4.10** ± 0.05|
> |SVHN     | 3.82 ± 0.06 | 3.47 ± 0.08 |**3.13** ± 0.17|**3.13** ± 0.14|
>
>
>
> These experiments demonstrate the efficacy of IAM across diverse and large-scale dataset, despite being conducted without hyper-parameter tuning under limited computational resources.
>
> ---
> > W3, Q2. Regarding the computational cost for the improvement of IAM.
>
> A clear discussion of the training cost is valuable to be added in the paper. We propose following analysis to address this point.
>
> **Computational Cost and Efficiency**
>
> As mentioned in line 249-250, our experiments are conducted with K=1 for Algorithm 1 for efficiency. This results in IAM requiring three forward passes and two backward passes per update step. Therefore, IAM's theoretical cost is approximately 2.5x that of standard SGD, while that of SAM is 2.0x.
>
>
> **Table**: Overhead comparison between optimization algorithm.
> |Optimizer|# of forward operations|# of backward operations|forward/backward time (cifar 10)|total cost (vs. SGD)|runnig time on ImageNet|
> |-|-|-|-|-|-|
> |SGD| 1| 1|31.33, 49.41 ms/step|  1.0x|533  s/epoch|
> |SAM| 2| 2|51.26, 95.48 ms/step| ~2.0x|-|
> |IAM| 3| 2|83.63, 99.10 ms/step| ~2.5x|1051 s/epoch|
>
> It's important to note the nature of the additional forward pass in IAM(-S). This pass is used to compute the reference model output, $f(θ)$, for the KL divergence term. Since it is not directly part of the backpropagation path for the main parameter update, it can be executed without storing intermediate activations. This makes the practical runtime overhead of IAM less substantial than the raw pass count might suggest, bringing its efficiency closer to that of SAM.
>
> Although IAM has a slightly higher theoretical cost than SAM, this is justified by its significant performance gain, particularly in challenging scenarios.
>
> On CIFAR-100 dataset, which is much more complex, IAM-S achieves a notable 0.75% reduction in test error over SAM, demonstrating its enhanced efficacy on difficult tasks.
>
> Moreover, the primary strength of IAM lies in its broad applicability to label-scarce tasks. The core advantage of our proposed local inconsistency measure is its ability to be computed only on unlabeled data. This is a fundamental difference from sharpness measures like that of SAM, which require labels. Therefore, the additional computational cost is a worthwhile trade-off for achieving robust performance in these practical, data-limited scenarios.
>
> ---
> >Q3. In semi-supervised learning (Table 2), why does SAM outperform IAM-D on CIFAR-10 at the 20% label rate?
>
> Our semi-SL experiments were designed to demonstrate the ability and unique advantage of IAM: its ability to leverage unlabeled data through its inherent label-free structure. Because the second-order geometry could be similar when the label data is sufficient, the benefit using second-order information of unlabeled data, could be smaller.
>
> To address concern about efficacy of IAM in semi-supervised setting, we have conducted a new experiment integrating our IAM-D with **FixMatch**, a powerful and widely adopted SSL framework that utilizes consistency regularization and pseudo-labeling. We simply added the local inconsistency term from IAM-D as an additional regularizer to standard FixMatch objective.
>
> **Table**: The results on CIFAR-10, using a WRN-28-10 model with only 250 labels (0.5%), are presented below:
> | Method|Test Error (%)|
> |-|-|
> |FixMatch (Baseline)|6.26 ± 0.39|
> |**FixMatch + IAM-D**|**5.30 ± 0.08**|
>
> This new result clearly shows that IAM-D provides a significant performance improvement when combined with a strong SSL baseline. It demonstrates that IAM serves as an effective and complementary **"plug-and-play" regularizer**. It enhances modern SSL frameworks by incorporating a principled geometric constraint, allowing the model to leverage second-order information from abundant unlabeled data in a way that existing methods do not.
>
> ---
> >Q4. Algorithm 1 initializes randomly, how does the choice of $\sigma^2$ affect the performance of IAM?
>
> $\sigma^2$ barely affects the performance of IAM. The estimation result of algorithm 1 hardly changes within sufficiently small $\sigma^2 \in [0.01, 10]$. As long as the quadratic approximation holds, what matters is the direction of the initial vector, not its magnitude.
>
>
>
> References:
>
> Sagun et al. (2017) Empirical analysis of the hessian of over-parametrized neural networks.
>
> Fort & Ganguli (2019) Emergent properties of the local geometry of neural loss landscapes.
>
> Luo et al. (2024) Explicit eigenvalue regularization improves sharpness-aware minimization.

---

> > ### Comment · Reviewer_5X7Q · 2025-08-05
> >
> > Thanks for the detailed response. My main concerns, including the formal version of Theorem 1 and its corresponding assumption,  experiments on large-scale datasets, and running time costs, are all addressed. The authors provide a more formal version of Theorem 1 and verify the effectiveness of the proposed method on large datasets like ImageNet. Besides, they also verify that the running time of their method is comparable to SAM. Currently, I have no further question, and I decide to retain my positive score.

---

### Official Review · Reviewer_XgQ6 · 2025-07-06

**Clarity:** 4
**Significance:** 2
**Originality:** 2
**Rating:** 4
**Confidence:** 4

**Summary:**

This paper introduces _local inconsistency_, a new notion for measuring the generalization error of trained models. Interestingly, local inconsistency does not depend on label data, making this notion usable for self-supervised and semi-supervised learning settings. The authors discuss connections between this notion and the maximum eigenvalue of the Fisher Information Matrix (FIM). They further present a PAC-Bayes generalization bound that depends on local inconsistency. Building on this result, they propose a fast method to approximate local inconsistency inspired by power methods. The authors introduce two new algorithms, IAM-D and IAM-S, which aim to minimize local inconsistency in addition to the empirical risk. In the end, the authors empirically show how consistently local inconsistency is correlated with training error, and also demonstrate the competitive performance of IAM-D and IAM-S on image datasets.

**Questions:**

- What is $m$ in equation 5?

**Ethical Concerns:**

["NO or VERY MINOR ethics concerns only"]

**Final Justification:**

I had some questions on # of gradient valuations; they got addressed, but experimental baselines in the paper are a little bit outdated. I hope authors address this in the final version as well.

**Limitations:**

Not discussed—especially, a discussion on the additional runtime and extra gradient valuations required for estimating local inconsistency would be very helpful.

**Paper Formatting Concerns:**

None.

**Quality:**

3

**Strengths And Weaknesses:**

Strengths:
- While the notion of local inconsistency is fairly simple, the experiments show that it is more effective in correlating with generalization error compared to the trace and maximum eigenvalue of the Hessian, especially in large models.
- The paper is very well written and easy to follow. The authors discuss the connection between local inconsistency, Fisher information, and inconsistency.

Weakness:
- Despite the multiple levels of approximation, estimating local inconsistency—and hence each iteration of the IAM-D and IAM-S algorithms—still requires $K$ additional gradient evaluations, which is computationally costly.
- Since local inconsistency and effectiveness of IAM-D and IAM-S are supported mostly experimentally, I would recommend a broader comparison to more recent SAM methods, for example, Li et al. [2024], Tahmasebi et al. [2024], Kwon et al. [2021], Mi et al. [2022], Mordido et. al. [2024] and others.
- The empirical improvements by IAM-D and IAM-S compared to baselines such as SAM or SGD are very minor. Given that IAM-D and IAM-S require $K$ additional gradient evaluations per iteration for estimating local inconsistency, while SAM and SGD require two and one respectively, it seems that if we allocate the same budget (in terms of gradient evaluations) and train baselines such as SGD and SAM longer, the improvement would be negligible. I recommend that the authors account for gradient evaluations in their experiments and report results using the same number of evaluations for the different methods.
- Tables demonstrating the additional runtime overhead of IAM-D and IAM-S are also quite helpful.

While I like the simple yet effective idea of this paper, given that the nature of the work is primarily experimental, I would, for now, lean toward rejection and reconsider later if my suggestions are addressed.

Minor Suggestion:
- It seems in equation for it should be $L_{S}(\theta)$.
- Typo: S_p(\theta), caption of subfig 1 of figure 1.
- Typo: Line 248, $L_{ISAM-D}(\theta)$.

References: \
Li et. al. Tilted Sharpness-Aware Minimization, ICML 2025, ArXIv 2024 \
Tahmasebi et. al. A Universal Class of Sharpness-Aware Minimization Algorithms, ICML 2024 \
Kwon et al. Asam: Adaptive sharpness-aware minimization for scale-invariant learning of deep neural networks, ICML 2021 \
Mi et al, Make sharpness-aware minimization stronger: A sparsified perturbation approach, NeurIPS 2022 \
Mordido  et. al. Lookbehind-SAM: k steps back, 1 step forward, ICML 2024

---

> ### Author Rebuttal · Authors · 2025-07-30
>
> > W1, W3, W4. Regarding concern about computational cost for IAM and evaluation.
>
> > Despite the multiple levels of approximation, estimating local inconsistency—and hence each iteration of the IAM-D and IAM-S algorithms—still requires $K$ additional gradient evaluations, which is computationally costly.
>
> >The empirical improvements by IAM-D and IAM-S compared to baselines such as SAM or SGD are very minor. Given that IAM-D and IAM-S require  additional gradient evaluations per iteration for estimating local inconsistency, while SAM and SGD require two and one respectively, it seems that if we allocate the same budget (in terms of gradient evaluations) and train baselines such as SGD and SAM longer, the improvement would be negligible. I recommend that the authors account for gradient evaluations in their experiments and report results using the same number of evaluations for the different methods.
>
> >Tables demonstrating the additional runtime overhead of IAM-D and IAM-S are also quite helpful.
>
> **Computational Cost and Efficiency**
> - IAM is **computationally efficient** since we used a **single  ascent step** $K=1$ in practice.
> - IAM **doesn't need additional gradient evaluation** (backward pass) compared to SAM.
> So we can allocate the same budget.
> - IAM shows better generalization than SGD with the same budget (trained **2.5x** longer).
>
> Our experiments are conducted with $K=1$ for Algorithm 1 for efficiency, as mentioned in line 249-250
> It is true that Algorithm 1 needs $K$ additional gradient evaluation.
> However, empirically local inconsistency almost converges when $K=3$ (please see figure in md file or figures folder in supplementary metrial). Moreover, $S_\rho(\theta)$ estimated with $K=3$ and $K=1$ are highly correlated with Pearson correlation coefficient above 0.9 in trained models for figure 1.
>
> Using $K=1$, IAM requires three forward passes and two backward passes (gradient evaluation) per update step. Therefore, IAM's theoretical cost is approximately 2.5x that of standard SGD, while SAM's is 2x of the SGD.
>
> **Table**: Overhead comparison between optimization algorithms.
> |Optimizer|# of forward operations|# of backward operations|CIFAR-10 forward/backward time (ms/step)|total cost (vs. SGD)|runnig time on ImageNet|
> |-|-|-|-|-|-|
> |SGD| 1| 1|31/49|  1.0x|533  s/epoch|
> |SAM| 2| 2|51/95| ~2.0x|-|
> |IAM| 3| 2|84/99| ~2.5x|1051 s/epoch|
>
> It's important to note a nature of the additional forward pass in IAM(-S). This pass is used to compute the reference model output, $f(θ)$ for the KL divergence term. Since it is not directly a part of the backpropagation path for the main parameter update, it can be executed without storing intermediate activations. This makes the practical runtime overhead of IAM less substantial than the raw pass count might suggest, bringing its efficiency closer to that of SAM.
>
> SGD does not show better generalization than IAM when trained twice longer in our ImageNet experiment with SGD and IAM-S. In many other papers such as Foret et al. [2021], Kwon et al. [2021], SGD shows no better generalization than SAM when trained twice longer.
>
> **Table**:  Top-{1, 5} error (mean ± stderr) (ImageNet) of IAM-S and SGD trained with **the same budget**.
> |Model    |error|IAM-S (200 epochs)| SGD (x2) | SGD (x2.5)|
> |---------|-----|----------------|-|-|
> |ResNet-50|Top-1|**21.90** ± 0.04|23.04 ± 0.08| 23.21|
> |         |Top-5|**5.98** ± 0.11 | 6.78 ± 0.02| 6.89|
>
> ---
> >W2. Since local inconsistency and effectiveness of IAM-D and IAM-S are supported mostly experimentally, I would recommend a broader comparison to more recent SAM methods.
>
> We compare our method to ASAM
> (Adaptive Sharpness-Aware Minimization) [Kwon et al. 2021] and IAM demonstrates performance comparable to, and often exceeding, that of ASAM.
>
> Table: Test Error (mean ± stderr) of different optimization methods on various datasets
> |Dataset|SGD|SAM|ASAM|IAM-D|IAM-S|
> |-|-|-|-|-|-|
> |CIFAR-10|3.95 ± 0.05|3.31 ± 0.01|**3.15** ± 0.02|3.28 ± 0.06|3.28 ± 0.03|
> |CIFAR-100|19.17 ± 0.19|17.63 ± 0.12|17.15 ± 0.11|17.16 ± 0.03|**16.82** ± 0.01|
> |F-MNIST|4.45 ± 0.05|4.13 ± 0.02|4.11 ±<0.01|4.13 ± 0.04|**4.10** ±<0.01|
> |SVHN|3.82 ± 0.06|3.47 ± 0.08|3.24 ± 0.04|**3.13** ± 0.05|**3.13** ± 0.01|
>
>
> Table: Test Error (mean ± stderr) of IAM-D, SAM, and SGD on WRN-16-8 trained with CIFAR-100 with varying label rates.
> |CIFAR-100|Model|1%|5%|10%|20%|
> |-|-|-|-|-|-|
> ||SGD|89.35 ± 0.098|72.65 ± 0.519|60.86 ± 0.204|50.01 ± 0.299|
> ||ASAM|89.63 ± 0.129|71.60 ± 0.377|57.84 ± 0.225|45.33 ± 0.165|
> ||IAM-D|**88.36 ± 0.292**|**69.99 ± 0.649**|**57.60 ± 0.251**|**45.05 ± 0.956**|
>
> Although the primary strength of IAM lies in its broad applicability to label-scarce tasks, The results show that our proposed IAM performs comparably to ASAM in the standard supervised learning setting. On the more complex CIFAR-100 dataset, IAM-S achieves superior generalization performance over ASAM.
>
> ---
> >Q1. What is $m$ in equation 5?
>
> $m$ is the number of parameters $\theta \in \mathbb{R}^{m}$.
>
> **We thank the reviewer for the thorough feedback, and we will incorporate all points discussed here into the forthcoming revision of the paper.**

---

> ### Comment · Reviewer_XgQ6 · 2025-08-02
>
> Thank you for the additional clarification on the number of gradient evaluations. I will raise my score, **provided that the authors include comparisons with more recent SAM methods in the camera‐ready version of the paper**. SAM is a vibrant, fast-paced field, and claiming competitive or state-of-the-art performance requires fair, up-to-date baselines. Although I’m not a fan of this narrative, when the main contributions are primarily experimental without significantly novel ideas, such comparisons are necessary to support the claims.

---

> > ### Author Response · Authors · 2025-08-09
> >
> > Thank you very much for your positive response.
> >
> > We confirm that we will, as you suggested, include the comparison experiments with more recent SAM methods, in the revision of the paper.
> >
> > We thank you once again for your time and insightful review.

---

### Decision · Program_Chairs · 2025-09-17

**Decision:**

Reject

**Comment:**

This paper introduces local inconsistency, a new measure of generalization error that does not require labels, making it suitable for self- and semi-supervised settings. The authors connect it to the maximum eigenvalue of the Fisher Information Matrix, derive a PAC-Bayes bound based on it, propose a fast approximation method, and introduce two algorithms (IAM-D and IAM-S) that minimize local inconsistency alongside empirical risk.

The reviewers were generally positive about the motivation and potential of the paper, with the exception of one, but a number of significant concerns emerged during the discussion. The proposed notion of local inconsistency appears to be a relatively incremental extension of the disagreement metric introduced by Jiang et al. and Johnson & Zhang, and the new algorithm is essentially a combination of two existing methods. On the empirical side, the improvements reported in Tables 1 and 2 are modest, despite the access to additional unlabeled data. The further experiments added during the rebuttal helped clarify some points but did not substantially strengthen the empirical case. A key limitation is the choice of baselines: only SGD and SAM are considered, while standard semi-supervised learning baselines are absent. This omission makes it difficult to assess the true advantages of the proposed approach. Taken together, the lack of clear novelty and the limited empirical validation weaken the contribution. While the paper may contain ideas worth exploring further, in its current form it does not provide sufficient evidence to support acceptance. Therefore, I recommend rejection, a recommendation the SAC agrees with.